# Learning interaction rules from multi-animal trajectories via augmented behavioral models

**Keisuke Fujii**[*]
Nagoya University
RIKEN Center for Advanced Intelligence Project
JST PRESTO

**Naoya Takeishi**
University of Applied Sciences
and Arts Western Switzerland
RIKEN Center for Advanced
Intelligence Project

**Kazushi Tsutsui**
Nagoya University

**Emyo Fujioka**
Doshisha University

**Nozomi Nishiumi**
National Institute
for Basic Biology

**Ryoya Tanaka**
Nagoya University

**Mika Fukushiro**
Doshisha University

**Kaoru Ide**
Doshisha University

**Hiroyoshi Kohno**
Tokai University

**Ken Yoda**
Nagoya University

**Susumu Takahashi**
Doshisha University

**Shizuko Hiryu**
Doshisha University

**Yoshinobu Kawahara**
Kyushu University
RIKEN Center for Advanced Intelligence Project

## Abstract

Extracting the interaction rules of biological agents from movement sequences pose challenges in various domains. Granger causality is a practical framework for analyzing the interactions from observed time-series data; however, this framework ignores the structures and assumptions of the generative process in animal behaviors, which may lead to interpretational problems and sometimes erroneous assessments of causality. In this paper, we propose a new framework for learning Granger causality from multi-animal trajectories via augmented theory-based behavioral models with interpretable data-driven models. We adopt an approach for augmenting incomplete multi-agent behavioral models described by time-varying dynamical systems with neural networks. For efficient and interpretable learning, our model leverages theory-based architectures separating navigation and motion processes, and the theory-guided regularization for reliable behavioral modeling. This can provide interpretable signs of Granger-causal effects over time, i.e., when specific others cause the approach or separation. In experiments using synthetic datasets, our method achieved better performance than various baselines. We then analyzed multi-animal datasets of mice, flies, birds, and bats, which verified our method and obtained novel biological insights.

## 1 Introduction

Extracting the interaction rules of real-world agents from data is a fundamental problem in a variety of scientific and engineering fields. For example, animals, vehicles, and pedestrians observe other's states and execute their actions in complex situations. Discovering the directed interaction rules of such agents from observed data will contribute to the understanding of the principles of biological

---

[*]fujii@i.nagoya-u.ac.jp

35th Conference on Neural Information Processing Systems (NeurIPS 2021).

agents' behaviors. Among methods analyzing directed interactions within multivariate time series, Granger causality (GC) [23] is a practical framework for exploratory analysis [49] in various fields, such as neuroscience [65] and economics [3] (see Section 2). Recent methodological developments have focused on inferring GC under nonlinear dynamics (e.g., [73, 31, 81, 55, 43]).

However, the structure of the generative process in biological multi-agent trajectories, which include navigational and motion processes [54] regarded as time-varying dynamical systems (see Section 3.1), is not fully utilized in existing base models of GC including vector autoregressive [27] and recent neural models [73, 31, 81]. Ignoring the structures of such processes in animal behaviors will lead to interpretational problems and sometimes erroneous assessments of causality. That is, incorporating the structures into the base model for inferring GC, e.g., augmenting (inherently) incomplete behavioral models with interpretable data-driven models (see Section 3.2), can solve these problems. Furthermore, since data-driven models sometimes detect false causality that is counterintuitive to the user of the analysis, e.g., introducing architectures and regularization to utilize scientific knowledge (see Sections 3.2 and 4.2) will be effective for a reliable base model of a GC method.

In this paper, we propose a framework for learning GC from biological multi-agent trajectories via augmented behavioral models (ABM) using interpretable data-driven neural models. We adopt an approach for augmenting incomplete multi-agent behavioral models described by time-varying dynamical systems with neural networks (see Section 3.2). The ABM leverages theory-based architectures separating navigation and motion processes based on a well-known conceptual behavioral model [54], and the theory-guided regularization (see Section 4.2) for interpretable and reliable behavioral modeling. This framework can provide interpretable signs of Granger-causal effects over time, e.g., when specific others cause the approach or separation.

The main contributions of this paper are as follows. (1) We propose a framework for learning Granger causality via ABM, which can extract interaction rules from real-world multi-agent and multi-dimensional trajectory data. (2) Methodologically, we realized the theory-guided regularization for reliable biological behavioral modeling for the first time. The theory-guided regularization can leverage scientific knowledge such that "when this situation occurs, it would be like this" (i.e., domain experts often know an input and output pair of the prediction model). Existing methods in Granger causality did not consider the utilization of such knowledge. (3) Biologically, our methodological contributions lies in the reformulation of a well-known conceptual behavioral model [54], which did not have a numerically computable form, such that we can compute and quantitatively evaluate it. (4) In the experiments, our method achieved better performance than various baselines using synthetic datasets, and obtained new biological insights and verified our method using multiple datasets of mice, birds, bats, and flies. In the remainder of this paper, we describe the background of GC in Section 2. Next, we formulate our ABM in Section 3, and the learning and inference methods in Section 4.

## 2 Granger Causality

GC [23] is one of the most popular and practical approaches to infer directed causal relations from observational multivariate time series data. Although the classical GC is defined by linear models, here we introduce a more recent definition of [73] for non-linear GC. Consider $p$ stationary time-series $\boldsymbol{x} = \{\boldsymbol{x}^1, ...\boldsymbol{x}^p\}$ across timesteps $t = \{1, ..., T\}$ and a non-linear autoregressive function $g_j$, such that

$$\boldsymbol{x}^j_{t+1} = g_j(\boldsymbol{x}^1_{\leq t}, ..., \boldsymbol{x}^p_{\leq t}) + \boldsymbol{\varepsilon}^j_t, \tag{1}$$

where $\boldsymbol{x}^j_{\leq t} = (..., \boldsymbol{x}^j_{t-1}, \boldsymbol{x}^j_t)$ denotes the present and past of series $j$ and $\boldsymbol{\varepsilon}^j_t$ represents independent noise. We then consider that variable $\boldsymbol{x}^i$ does not Granger-cause variable $\boldsymbol{x}^j$, denoted as $\boldsymbol{x}^i \nrightarrow \boldsymbol{x}^j$, if and only if $g_j(\cdot)$ is constant in $\boldsymbol{x}^i_{\leq t}$. Granger causal relations are equivalent to causal relations in the underlying directed acyclic graph if all relevant variables are observed and no instantaneous (i.e., connections between two variables at the same timestep) connections exist [59]. Many methods for Granger causal discovery, including vector autoregressive [27] and recent deep learning-based approaches [73, 31, 81], can be encapsulated by the following framework. First, we define a function $f_\theta$ (e.g., an multilayer perceptrons (MLP) in [73], a linear model in [27]), which learns to predict the next time-step of the test sequence $\boldsymbol{x}$. Then, we fit $f_\theta$ to $\boldsymbol{x}$ by minimizing some loss (e.g., mean squared error) $\mathcal{L}$: $\theta_\star = \mathrm{argmin}_\theta \mathcal{L}(\boldsymbol{x}, f_\theta)$. Finally, we apply some fixed function $h$ (e.g., thresholding) (e.g., [45]) to the learned parameters to produce a Granger causal graph estimate for $\boldsymbol{x}$: $\hat{\mathcal{G}}_{\boldsymbol{x}} = h(\theta_\star)$.

Furthermore, we need to differentiate between positive and negative Granger-causal effects (e.g, approaching and separating). Based on the definition of [45], we define the effect sign as follows: if $g_j(\cdot)$ is increasing in all $x^i_{\leq t}$, then we say that variable $x^i$ has a positive effect on $x^j$, if $g_j(\cdot)$ is decreasing in $x^i_{\leq t}$, then $x^i$ has a negative effect on $x^j$. Note that $x^i$ can contribute both positively and negatively to the future of $x^j$ at different delays.

Overall, the causality measures, however elaborate in construction, are simply statistics estimated from a model [71]. If the model inadequately represents the system properties of interest, subsequent analyses based on the model will fail to address the question of interest. The inability of the model to represent key features of interest can cause interpretational problems and sometimes erroneous assessments of causality. Therefore, in our case, incorporating the structures of the generative process for animal behaviors (i.e., Eq.(2)) in a numerically computable form will be required. We thus propose the ABM based on a well-known conceptual model [54] in biological sciences in the next section.

## 3 Augmented behavioral model

Our motivation for developing interpretable behavior models is to obtain new insights from the results of Granger causality. In this section, we firstly formulate a well-known conceptual behavioral model [54] so that it can be computable. Second, we propose (multi-animal) ABMs with theory-based architectures based on scientific knowledge. Further, we discuss the relation to the existing explainable neural models [2]. The diagram of our method is described in Appendix C.

### 3.1 Formulation of a conceptual behavioral model

In movement ecology, which is a branch of biology concerning the spatial and temporal patterns of behaviors of organisms, a coherent framework [54] has been conceptualized to explore the causes, mechanisms, and patterns of movement. For example, two alternative structural representations [54] were proposed to model a new position $p_{t+1}$ from its current location $p_t$ (for details, see Appendix A): the motion-driven case $p_{t+1} = f_U(f_M(\mathbf{\Omega}, f_N(\mathbf{\Phi}, r_t, w_t, p_t), r_t, w_t, p_t)) + \varepsilon_t$, and the navigation-driven case $p_{t+1} = f_U(f_N(\mathbf{\Phi}, f_M(\mathbf{\Omega}, r_t, w_t, p_t), r_t, w_t, p_t)) + \varepsilon_t$, where $w_t$ is the internal state, $\mathbf{\Omega}$ is the motion capacity, $\mathbf{\Phi}$ is the navigation capacity, and $r_t$ is the environmental factors (these are conceptual parameters). $f_M$, $f_N$, and $f_U$ are conceptual functions to represent actions of the motion (or planning), navigation, and movement progression processes, respectively.

For efficient learning of the weights in the model (i.e., coefficient of Granger causality) in this paper, we consider a simple case with homogeneous navigation and motion capacities, and internal states. Moreover, to make the contribution of $f_M$, $f_N$, and $f_U$ interpretable after training from the data for extracting unknown interaction rules (and for obtaining scientific new insights), one of the simplified processes for agent $i$ is represented by

$$x^i_{t+1} = f^i_U(f^i_N(r^i_t, x^i_t), f^i_M(r^i_t, x^i_t), r^i_t, x^i_t) + \varepsilon^i_t, \tag{2}$$

where $x^i \in \mathbb{R}^d$ includes location $p^i$ and velocity for the agent $i$. We here consider $r^i \in \mathbb{R}^{(p-1)d_r}$ including $p-1$ other agents' $d_r$-dimensional information. This formulation does not assume either motion-driven or navigation-driven case. Such behaviors have been conventionally modeled by mathematical equations such as force- and rule-based models (e.g., reviewed by [77, 47]). Recently, these models have become more sophisticated by incorporating the models into hand-crafted functions representing anticipation (e.g., [29, 51]) and navigation (e.g., [8, 76]).

However, these conventional and recent models are sometimes too simplistic and customized for the specific animals, respectively; thus it is sometimes difficult to define the dynamics of general biological multi-agent systems (i.e., multiple species of animals). Therefore, methods for learning parameters and interaction rules of behavioral models are needed. There have been some researches to estimate specific parameters (and their distributions) of the interpretable behavior models (e.g., [84, 85, 15]), and others to model the parameters and rules in purely data-driven manners (i.e., sometimes uninterpretable) only for accurate prediction (e.g., [16, 28]). In the proposed framework, we consider flexible data-driven interpretable models to focus on inferring GC for exploratory analysis from the observed data without specific knowledge of the species and obtaining additional data.

Recently, some attempts have been made to explore flexible and interpretable models bridging theory-based and data-driven approaches. For example, a paradigm called theory-guided data science has

been proposed [30], which leverages the wealth of scientific knowledge for improving the effectiveness of data-driven models in enabling scientific discovery. For example, scientific knowledge can be used as architectures or regularization terms in learning algorithms in physical and biological sciences (e.g., [61, 21]). In biological multi-agent systems, an approach extract interpretable dynamical information based on physics-based knowledge [18] from multi-agent interaction data, and another approach made a particular module such as observation (e.g., [19]) interpretable in mostly black-box neural models. However, these data-driven models did not sufficiently utilize the above scientific knowledge of multi-animal interactions. In the next subsection, to make the model (e.g., of GC) flexible and interpretable, we propose an ABM with theory-based architectures.

## 3.2  Augmented behavioral model with theory-based architectures

In this subsection, we propose a ABM using interpretable neural models with theory-based architectures for learning GC from multi-animal trajectories. In general, it is scientifically beneficial if a model mimics the data-generating process well, e.g., because existing scientific insights can be leveraged or revalidated. In our case of GC, additionally, it is expected to eliminate obvious erroneous causality by utilizing existing knowledge, we thus propose a theory-based ABM for learning GC.

Generally, scientific knowledge can be used to influence the architecture of data-driven scientific models. Most design considerations are mainly motivated to simplify the learning procedure, minimize the training loss, and ensure robust generalization performance [30]. In some cases, domain knowledge can be used designing neural models by decomposing the overall problem into modular sub-problems. For example, in our problem, to describe the overall process of multi-animal behaviors, modular neural models can be learned for different sub-processes, such as the navigation, planning, and movement processes ($f_N^i, f_M^i$, and $f_U^i$, respectively) described in Section 3.1. This will help in using the power of learning frameworks while following a high-level organization in the architecture that is motivated by domain knowledge [30]. Specifically, to accurately model the relationships between agents (finally interpreted as causal relationships) with limited information in usual GC settings, we explicitly formulate the $f_N^i, f_M^i$, and $f_U^i$ and estimate $f_N^i$ and $f_M^i$ from data.

In summary, our base ABM can be expressed as

$$\boldsymbol{x}_t^i = \sum_{k=1}^{K} \left( F_N^{i,t,k}(\boldsymbol{h}_{t-k}^i) \odot F_M^{i,t,k}(\boldsymbol{h}_{t-k}^i) \right) \boldsymbol{h}_{t-k}^i + \boldsymbol{\varepsilon}_t^i, \tag{3}$$

where $\boldsymbol{h}_{t-k}^i \in \mathbb{R}^{d_h}$ is a vector concatenating the self state $\boldsymbol{x}_{t-k}^i \in \mathbb{R}^d$ and all others' state $\boldsymbol{r}_{t-k}^i \in \mathbb{R}^{(p-1)d_r}$, and $\odot$ denotes a element-wise multiplication. $K$ is the order of the autoregressive model. $F_N^{i,t,k}, F_M^{i,t,k} : \mathbb{R}^{d_h} \to \mathbb{R}^{d \times d_h}$ are matrix-valued functions that represent navigation and motion functions, which are implemented by MLPs. For brevity, we omit the intercept term here and in the following equations. The value of the element of $F_N^{i,k}$ is $[-1, 1]$ is like a switching function value, i.e., a positive or negative sign to represent the approach and separation from others. The value of the element of $F_M^{i,k}$ is a positive value or zero, which changes continuously and represents coefficients of time-varying dynamics. Relationships between agents $\boldsymbol{x}^1, ..., \boldsymbol{x}^p$ and their variability throughout time can be examined by inspecting coefficient matrices $\boldsymbol{\Psi}_{\boldsymbol{\theta}_{t,k}}^i = \left( F_N^{i,t,k}(\boldsymbol{h}_{t-k}^i) \odot F_M^{i,t,k}(\boldsymbol{h}_{t-k}^i) \right)$. We separate $\boldsymbol{\Psi}_{\boldsymbol{\theta}_{t,k}}^i$ into $F_N^{i,t,k}(\boldsymbol{h}_{t-k}^i)$ and $F_M^{i,t,k}(\boldsymbol{h}_{t-k}^i)$ for two reasons: interpretability and efficient use of scientific knowledge. The interpretability of two coefficients $F_N^{i,k}$ and $F_M^{i,k}$ contributes to the understanding of navigation and motion planning processes of animals (i.e., signs and amplitudes in the GC effects), respectively. The efficient use of scientific knowledge in the learning of a model enables us to incorporate the knowledge into the model. The effectiveness was shown in the ablation studies in the experiments. Specific forms of Eq. (3) are described in Appendices E.2 and G.2. The formulation of the model via linear combinations of the interpretable feature $\boldsymbol{h}_{t-k}^i$ for an explainable neural model is related to the self-explanatory neural network (SENN) [2].

## 3.3  Relation to self-explanatory neural network

SENN [2] was introduced as a class of intrinsically interpretable models motivated by explicitness, faithfulness, and stability properties. A SENN with a link function $g(\cdot)$ and interpretable basis concepts $h(\boldsymbol{x}) : \mathbb{R}^p \to \mathbb{R}^u$ follows the form

$$f(\boldsymbol{x}) = g(\theta(\boldsymbol{x})_1 h(\boldsymbol{x})_1, ..., \theta(\boldsymbol{x})_u h(\boldsymbol{x})_u), \tag{4}$$

where $\boldsymbol{x} \in \mathbb{R}^p$ are predictors; and $\theta(\cdot)$ is a neural network with $u$ outputs (here, we consider the simple case of $d = 1$ and $d_r = 1$). We refer to $\theta(\boldsymbol{x})$ as generalized coefficients for data point $\boldsymbol{x}$ and use them to *explain* contributions of individual basis concepts to predictions. In the case of $g(\cdot)$ being sum and concepts being raw inputs, Eq. (4) simplifies to $f(\boldsymbol{x}) = \sum_{i=1}^p \theta(\boldsymbol{x})_i \boldsymbol{x}_i$. In this paper, we regard the movement function $f_U^i$ as $g(\cdot)$ and the function of $f_N^i$ and $f_M^i$ as $\theta$ for the following interpretable modeling of $f_U^i$, $f_N^i$, and $f_M^i$. Appendix B presents additional properties SENNs need to satisfy and the learning algorithm, as defined by [2]. Note that our model does not always satisfy the requirements of SENN [2, 45] due to the modeling of time-varying dynamics (see Appendix B). SENN was first applied to GC [45] via generalized vector autoregression model (GVAR): $\boldsymbol{x}_t = \sum_{k=1}^K \boldsymbol{\Psi}_{\boldsymbol{\theta}_k}(\boldsymbol{x}_{t-k}) \boldsymbol{x}_{t-k} + \boldsymbol{\varepsilon}_t$, where $\boldsymbol{\Psi}_{\boldsymbol{\theta}_k} : \mathbb{R}^p \to \mathbb{R}^{p \times p}$ is a neural network parameterized by $\boldsymbol{\theta}_k$. $\boldsymbol{\Psi}_{\boldsymbol{\theta}_k}(\boldsymbol{x}_{t-k})$ is a matrix whose components correspond to the generalized coefficients for lag $k$ at timestep $t$. The component $(i, j)$ of $\boldsymbol{\Psi}_{\boldsymbol{\theta}_k}(\boldsymbol{x}_{t-k})$ corresponds to the influence of $\boldsymbol{x}_{t-k}^j$ on $\boldsymbol{x}_t^i$. However, the SENN model did not use scientific knowledge of multi-element interactions and may cause interpretational problems and sometimes erroneous assessments of causality.

# 4 Learning with theory-guided regularization and inference

Here, we describe the learning method of the ABM including theory-guided regularization. We first overview the learning method and define the objective function. We then explain the theory-guided regularization for incorporating scientific knowledge into the learning of the model. Finally, we describe the inference of GC by our method. Again, the overview of our method is described in Appendix C.

## 4.1 Overview

To mitigate the inference in multivariate time series, Eq. (3) for each agent is summarized as the following expression:

$$\boldsymbol{x}_t = \sum_{k=1}^K \left[ \left( F_N^{1,t,k}(\boldsymbol{h}_{t-k}^1) \odot F_M^{1,t,k}(\boldsymbol{h}_{t-k}^1) \right) \boldsymbol{h}_{t-k}^1, \dots, \left( F_N^{p,t,k}(\boldsymbol{h}_{t-k}^p) \odot F_M^{p,t,k}(\boldsymbol{h}_{t-k}^p) \right) \boldsymbol{h}_{t-k}^p \right] + \boldsymbol{\varepsilon}_t, \quad (5)$$

where $\boldsymbol{x}_t$ and $\boldsymbol{\varepsilon}_t$ concatenate the original variables for all $p$ agents (various $F$s and $Psi$s are learned parameters). We train our model by minimizing the following penalized loss function with the mini-batch gradient descent

$$\sum_{t=K+1}^T \left( \mathcal{L}_{pred}(\hat{\boldsymbol{x}}_t, \boldsymbol{x}_t) + \lambda \mathcal{L}_{sparsity}(\boldsymbol{\Psi}_t) + \gamma \mathcal{L}_{TG}(\boldsymbol{\Psi}_t, \boldsymbol{\Psi}_t^{TG}) \right) + \sum_{t=K+1}^{T-1} \beta \mathcal{L}_{smooth}(\boldsymbol{\Psi}_{t+1}, \boldsymbol{\Psi}_t), \quad (6)$$

where $\{\boldsymbol{x}_t\}_{t=1}^T$ is a single observed time series of length $T$ with $d$-dimensions and $p$-agents; $\hat{\boldsymbol{x}}_t$ is the one-step forecast for the $t$-th time point based on Eq. (5); $\boldsymbol{\Psi}_t \in \mathbb{R}^{pd \times Kd_h}$ is defined as a concatenated matrix of $[\boldsymbol{\Psi}_{\boldsymbol{\theta}_{t,K}}^1, \dots, \boldsymbol{\Psi}_{\boldsymbol{\theta}_{t,1}}^1], \dots, [\boldsymbol{\Psi}_{\boldsymbol{\theta}_{t,K}}^p, \dots, \boldsymbol{\Psi}_{\boldsymbol{\theta}_{t,1}}^p]$ in a row; $\boldsymbol{\Psi}_t^{TG}$ is a coefficient determined by the following theory-guided regularization; and $\lambda, \beta, \gamma \geq 0$ are regularization parameters. The loss function in Eq. (6) consists of four terms: (i) the mean squared error (MSE) prediction loss, (ii) a sparsity-inducing penalty term, (iii) theory-guided regularization, and (iv) the smoothing penalty term. The sparsity-inducing term $\mathcal{L}_{sparsity}$ is an appropriate penalty on the norm of $\boldsymbol{\Psi}_t$. Among possible various regularization terms, in our implementation, we employ the elastic-net-style penalty term [86, 56] $\mathcal{L}_{sparsity}(\boldsymbol{\Psi}_t) = \frac{1}{T-K} \left( \alpha \|\boldsymbol{\Psi}_t\|_1 + (1-\alpha) \|\boldsymbol{\Psi}_t\|_F^2 \right)$, with $\alpha = 0.5$, based on [45]. Note that other penalties can be also easily adapted to our model. The smoothing penalty term, given by $\mathcal{L}_{smooth}(\boldsymbol{\Psi}_{t+1}, \boldsymbol{\Psi}_t) = \frac{1}{T-K-1} \|\boldsymbol{\Psi}_{t+1} - \boldsymbol{\Psi}_t\|_F^2$, is the average norm of the difference between generalized coefficient matrices for two consecutive time points. This penalty term encourages smoothness in the evolution of coefficients with respect to time [45]. To avoid overfitting and model selection problems, we eliminate unused factors based on prior knowledge (for details, see Appendices E.2 and G.2).

## 4.2 Theory-guided regularization

The third term in Eq. (6) is the theory-guided regularization for reliable Granger causal discovery by leveraging regularization with scientific knowledge. Here we utilize theory-based and data-driven prediction results and impose penalties in the appropriate situations as described below. Again, let $\hat{\boldsymbol{x}}_t$ be the prediction from the data. In addition to the data, we prepare some input-output

pairs $(\tilde{\boldsymbol{x}}_{t-k\leq t}, \tilde{\boldsymbol{x}}_t)$ based on scientific knowledge. We call them pairs of theory-guided feature and prediction, respectively. In this case, we assume that the theory-guided cause or weight of the ABM $\boldsymbol{\Psi}_t^{TG}$ is uniquely determined. When the difference between $\hat{\boldsymbol{x}}_t$ and $\tilde{\boldsymbol{x}}_t$ is below a certain threshold, we assume that the cause (weight) of $\hat{\boldsymbol{x}}_t$ is equivalent to the cause of $\tilde{\boldsymbol{x}}_t$.

In animal behaviors, the theory-guided prediction utilizes the intuitive prior knowledge such that the agents go straight from the current state if there is no interaction. In this case, $\tilde{\boldsymbol{x}}_t$ includes the same velocity as the previous step and the corresponding positions after going straight. The penalty is expressed as $\mathcal{L}_{TG}(\boldsymbol{\Psi}_t, \boldsymbol{\Psi}_t^{TG}) = \frac{1}{T-K} \exp(\|\boldsymbol{x}_t - \tilde{\boldsymbol{x}}_t\|_2^2/\sigma)\|\boldsymbol{\Psi}_t'\|_F^2$, where $\boldsymbol{\Psi}_t' \in \mathbb{R}^{pd \times K(p-1)d_r}$ is the weight matrix regarding others' information (i.e., eliminating the information of the agents themselves from $\boldsymbol{\Psi}_t$) and $\sigma$ is a parameter regarding the threshold. Note that here the matrix $\boldsymbol{\Psi}_t'^{TG}$ corresponding to $\boldsymbol{\Psi}_t^{TG}$ is a zero matrix representing no interaction with others (i.e., $\|\boldsymbol{\Psi}_t' - \boldsymbol{\Psi}_t'^{TG}\|_F^2 = \|\boldsymbol{\Psi}_t'\|_F^2$).

Next, we can consider the general cases. All possible combinations of the pairs are denoted as the direct product $\mathcal{H}_0 := L \times M \times \cdots \times M = \{(l, m_1, \ldots, m_p) \mid l \in L \wedge m_1 \in M \wedge \cdots \wedge m_p \in M\}$, where $L = \{1, \ldots, p\}$ and $M = \{-1, 0, 1\}$ if we consider the sign of Granger causal effects (otherwise, $M = \{0, 1\}$). However, if we consider the pairs $(\tilde{\boldsymbol{x}}_{t-k\leq t}, \tilde{\boldsymbol{x}}_t)$ uniquely determined, it will be a considerably fewer number of combinations by avoiding underdetermined problems. We denote the set of the uniquely-determined combinations as $\mathcal{H}_1 \subset \mathcal{H}_0$. We can then impose penalties on the weights: $\mathcal{L}_{TG}(\boldsymbol{\Psi}_t, \boldsymbol{\Psi}_t^{TG}) = \frac{1}{|\mathcal{H}_1|(T-K)} \sum_{l,m_1,\ldots,m_p \in \mathcal{H}_1} \left( \exp(\|\boldsymbol{x}_t - \tilde{\boldsymbol{x}}_t\|_F^2/\sigma)\|\boldsymbol{\Psi}_t' - \boldsymbol{\Psi}_{t,l,m_1,\ldots,m_p}'^{TG}\|_F^2 \right)$, where $\boldsymbol{\Psi}_{t,l,m_1,\ldots,m_p}'^{TG} \in \mathbb{R}^{pd \times K(p-1)d_r}$ is the weight matrix regarding others' information in $\boldsymbol{\Psi}_t$. In animal behaviors, due to unknown terms, such as inertia and other biological factors, the theory-guided prediction utilizes the only intuitive prior knowledge such that the agents go straight from the current state if there are no interactions (i.e., $|\mathcal{H}_1| = 1$).

## 4.3 Inference of Granger causality

Once $\boldsymbol{\Psi}_t$ is trained, we quantify strengths of Granger-causal relationships between variables by aggregating matrices $\boldsymbol{\Psi}_t$ across all $K, d, d_r, t$ into summary statistics. Although most neural GC methods [73, 55, 31, 81, 45] did not provide an obvious way for handling multi-dimensional time series (i.e., $d > 1$), our main problems include two- or three-dimensional positional and velocity data for each animal. Therefore, we compute the norm with respect to spatial dimensions $d, d_r$, and the sign of the GC separately. That is, we aggregate the obtained generalized coefficients into matrix $\boldsymbol{S} \in \mathbb{R}^{p \times p}$ as follows:

$$S_{i,j} = \operatorname*{signmax}_{K+1 \leq t \leq T} \left\{ \operatorname*{signmax}_{1 \leq k \leq K} \left( \operatorname*{median}_{\substack{q=1,\ldots,d_r \\ u=1,\ldots,d}} (\boldsymbol{\Psi}_{i,j}) \right) \right\} \max_{K+1 \leq t \leq T} \left( \max_{1 \leq k \leq K} \left( \| (\boldsymbol{\Psi}_{i,j})_{t,k} \|_F \right) \right), \quad (7)$$

where $\boldsymbol{\Psi}_{i,j} \in \mathbb{R}^{(T-K) \times K \times d \times d_r}$ is computed by reshaping and concatenating $\boldsymbol{\Psi}_t$ over $K+1 \leq t \leq T$. $\| (\boldsymbol{\Psi}_{i,j})_{t,k} \|_F$ is the Frobenius norm of the matrix $(\boldsymbol{\Psi}_{i,j})_{t,k} \in \mathbb{R}^{d \times d_r}$ for each $t, k$. The *signmax* is an original function to output the sign of the larger value of the absolute value of the maximum and minimum values (e.g., signmax($\{1, 2, -3\}$) = $-1$). If we do not consider the sign of Granger causal effects, we ignore the coefficient of the signed function. If we investigate the GC effects over time, we eliminate max function among $t$. Note that we only consider off-diagonal elements of adjacency matrices and ignore self-causal relationships. Intuitively, $S_{i,j}$ are statistics that quantify the strength of the Granger-causal effect of $\boldsymbol{x}^i$ on $\boldsymbol{x}^j$ using magnitudes of generalized coefficients. We expect $S_{i,j}$ to be close to 0 for non-causal relationships and $S_{i,j} \gg 0$ if $\boldsymbol{x}^i \to \boldsymbol{x}^j$. Note that in practice $S_{i,j}$ is not binary-valued, as opposed to the ground truth, which we want to infer, because the outputs of $\boldsymbol{\Psi}_{i,j}$ are not shrunk to exact zeros. Therefore, we need a procedure deciding for which variable pairs $S_{i,j}$ are significantly different from 0. To infer a binary matrix of GC relationships, we use a heuristic threshold. For the detail, see Appendix D.1.

## 5 Related work

**Methods for nonlinear GC.** Initial work for nonlinear GC methods focused on time-varying dynamic Bayesian networks [70], regularized logistic regression with time-varying coefficients [34], and kernel-based regression models [46, 69, 39]. Recent approaches to inferring Granger-causal relationships leverage the expressive power of neural networks [50, 79, 73, 55, 31, 43, 81] and are often based on

regularized autoregressive models. Methods using sparse-input MLPs and long short-term memory to model nonlinear autoregressive relationships have been proposed [73], followed by a more sample efficient economy statistical recurrent unit (eSRU) architecture [31]. Other researchers proposed a temporal causal discovery framework that leverages attention-based convolutional neural networks [55] and a framework to interpret signs of GC effects and their variability through time building on SENN [2]. However, the structure of time-varying dynamical systems in multi-animal trajectories was not fully utilized in the above models.

**Information-theoretic analysis for multi-animal motions.** In this topic, most researchers have adopted transfer entropy (TE) and its variants and have analyzed them in terms of e.g., information cascades rather than causal discovery among animals. In the pioneering work, [78] analyzed information cascades among artificial collective motions using (conditional) TE [40, 41]. [64] applied variants of conditional mutual information to identify dynamical coupling between the trajectories of foraging meerkats. TE has been used to study the response of schools of zebrafish to a robotic replica of the animal [9, 36], to infer leadership in pairs of bats [58] and simulated zebrafish [10], and to identify interactions in a swarm of insects (Chironomus riparius) [42]. Local TE (or pointwise TE) [67, 40] has been used to detect local dependencies at specific time points in a swarm of soldier crabs [75], teams of simulated RoboCup agents [11], and a school of fish [13]. Since biological collective motions are intrinsically time-varying dynamical systems, we compared our methods and local TE in our experiments.

**Other Biological multi-agent motion analysis.** Previous studies have investigated leader-follower relationships. For example, the existences of the leadership have been investigated via the correlation in movement with time delay (e.g., [53, 66]) and via global physical (e.g., [4]) and statistical properties [57]. Meanwhile, methods for data-driven biological multi-agent motion modeling have been intensively investigated for pedestrian (e.g., [1, 25]), vehicles (e.g., [5, 63, 72]), animals [16, 28], and athletes (e.g., [83, 37]). In most of these methods, the agents are assumed to have the full observation of other agents to achieve accurate prediction. In contrast, some researches have modeled partial observation in real-world multi-agent systems [26, 38, 24, 17, 18, 19, 22]. However, the above approaches required a large amount of training data and would not be suitable for application to the multi-animal trajectory datasets that are measured in small quantities.

## 6 Experiments

The purpose of our experiments is to validate the proposed methods for application to real-world multi-animal movement trajectories, which have usually a smaller amount of sequences and no ground truth of the interaction rules. Thus, for verification of our methods, we first compared their performances to infer the Granger causality to those in various baselines using two synthetic datasets with ground truth: nonlinear oscillator (Kuramoto model) and boid model simulation datasets. We used the same ABM as applied to real-world multi-animal trajectory datasets: mice, birds, bats, and flies. To demonstrate the applicability to the multi-element dynamics other than multi-animal trajectories, we validated our method using the Kuramoto dataset (the results are shown in Appendix F). Each method was trained only on one sequence according to most neural GC frameworks [73, 31, 45]. The hyperparameters of the models were determined by validation datasets in the synthetic data experiments (for the details, see Appendices E and G). The common training details, (binary) inference methods, computational resources, and the amount of computation are described in Appendix D. Our code is available at `https://github.com/keisuke198619/ABM`.

### 6.1 Synthetic datasets

For verification of our method, we compared the performances to infer the GC to those in various baselines using two synthetic datasets with ground truth. To compare with various baselines of GC methods, we tackled problems where the true causality is not changed over time. Here, we compared our methods (ABM) to 5 baseline methods: economic statistical recurrent unit (eSRU) [31]; amortized causal discovery (ACD) [43]; GVAR [45] (this is the most appropriate baseline); and simple baselines such as linear GC and local TE modified from [81, 43]. Except for ACD [43], most baselines did not provide an obvious way for handling multi-dimensional time series, whereas our main problems include two- or three-dimensional trajectories for each animal. Therefore, we modified the baselines except for ACD to compute norms with respect to spatial dimensions (2 or 3) for comparability with the proposed method. Note that the interpretations of the relationships estimated by ACD and Local TE are different from other methods, thus the sign of the relationship could not be investigated (we denote such by N/A in Table 1).

We investigated the continuously-valued inference results: the norms of relevant weights, scores, and strengths of GC relationships. We compared these scores against the true structures using areas under receiver operating characteristic (AUROC) and precision-recall (AUPRC) curves. We also evaluated thresholded inference results: accuracy (Acc) and balanced accuracy (BA) scores. For the inference methods of a binary matrix of GC relationships, see Appendix D.1. For all evaluation metrics, we only considered off-diagonal elements of adjacency matrices, ignoring self-causal relationships.

**Boid model**. Here, we evaluated the interpretability and validity of our method on the simulation data using the boid model, which contains five agents movement trajectories (for details, see Appendix G). In this experiment, we set the boids (agents) directed preferences: we randomly set the ground truth relationships $1, 0$, and $-1$ as the rules of attraction, no interaction, and repulsion, respectively. Figure 1 illustrates that e.g., boid #5 was attracted to boid #1 (i.e., true relationship: 1) and boid #3 avoided boid #1 (true relationship: $-1$). In this figure, our method detected the changes in the signed relationships whereas the GVAR [45] did not.

The performances were evaluated using $S_{i,j}$ in Eq. (7) throughout time because our method and ground truth were sensitive the sign as shown in Figure 1. Table 1 (upper) shows that our method achieved better performance than various baselines. The ablation studies shown in Table 1 (lower) reveal that the main two contributions of this work, the theory-guided regularization $\mathcal{L}_{TG}$ and learning navigation function $\boldsymbol{F}_N^k$ and motion function $\boldsymbol{F}_M^k$ separately, improved the performance greatly. These suggest that the utilization of scientific knowledge via the regularization and architectures efficiently worked in the limited data situations. Similarly, the results of the Kuramoto dataset are shown in Appendix F, indicating that our method achieved much better performance than these baselines. Therefore, our method can effectively infer the GC in multi-agent (or multi-element) systems with partially known structures.

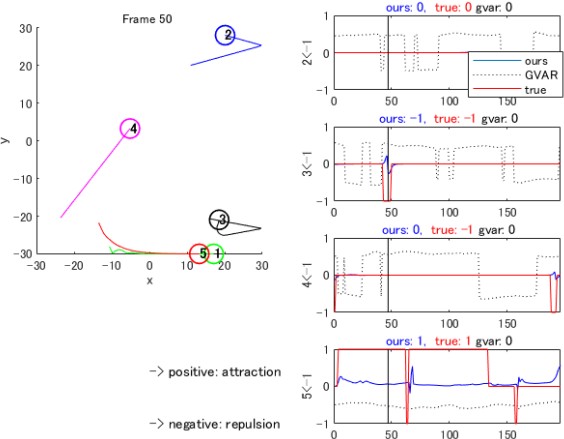

Figure 1: Example results of the boid model. Left: five boids (agents) movements. Trajectories are histories of the movement. Right: the results of our method (blue) and GVAR [45] (black dash) for the relationship between the cause (boid #1) and effects (other boids; i.e., $S_{i,j}$ for $i = 1$ and $j = 2, \ldots, 5$). The binary relationships are described in the upper of the plot. $1, 0$, and $-1$ indicate attraction, no interaction, and repulsion, respectively. Note that the magnitudes of our method and GVAR [45] were normalized with their maximal values (thus, the values were not be comparable among methods and red ground truth). For the detail, see the main text.

|  | Boid model | | | |
|---|---|---|---|---|
|  | Bal. Acc. | AUPRC | $BA_{pos}$ | $BA_{neg}$ |
| Linear GC | $0.487 \pm 0.028$ | $0.591 \pm 0.169$ | $0.55 \pm 0.150$ | $0.530 \pm 0.165$ |
| Local TE | $0.634 \pm 0.130$ | $0.580 \pm 0.141$ | N/A | N/A |
| eSRU [31] | $0.500 \pm 0.000$ | $0.452 \pm 0.166$ | $0.495 \pm 0.102$ | $0.508 \pm 0.153$ |
| ACD [43] | $0.411 \pm 0.099$ | $0.497 \pm 0.199$ | N/A | N/A |
| GVAR [45] | $0.441 \pm 0.090$ | $0.327 \pm 0.119$ | $0.524 \pm 0.199$ | $0.579 \pm 0.126$ |
| ABM - $\boldsymbol{F}_N$ - $\mathcal{L}_{TG}$ | $0.500 \pm 0.021$ | $0.417 \pm 0.115$ | $0.513 \pm 0.096$ | $0.619 \pm 0.157$ |
| ABM - $\boldsymbol{F}_N$ | $0.542 \pm 0.063$ | $0.385 \pm 0.122$ | $0.544 \pm 0.160$ | $0.508 \pm 0.147$ |
| ABM - $\mathcal{L}_{TG}$ | $0.683 \pm 0.124$ | $0.638 \pm 0.096$ | $0.716 \pm 0.172$ | $0.700 \pm 0.143$ |
| ABM (ours) | $\mathbf{0.767 \pm 0.146}$ | $\mathbf{0.819 \pm 0.126}$ | $\mathbf{0.724 \pm 0.189}$ | $\mathbf{0.760 \pm 0.160}$ |

Table 1: Performance comparison on the boid model.

## 6.2 Multi-animal trajectory datasets

We here analyzed biological multi-agent trajectory datasets of bats, birds, mice, and flies and obtained new biological insights using our framework (for the results of flies, see Appendix H). We used the same ABM as used in the boid dataset. In real-world data, since there was no ground truth, we used the hyperparameters of the boid simulation dataset. As a possible verification method, our method can be verified by investigating whether the GC result follows the hypothesis using mice and flies (Appendix H) datasets, which were controlled based on scientific knowledge. Next, we show that

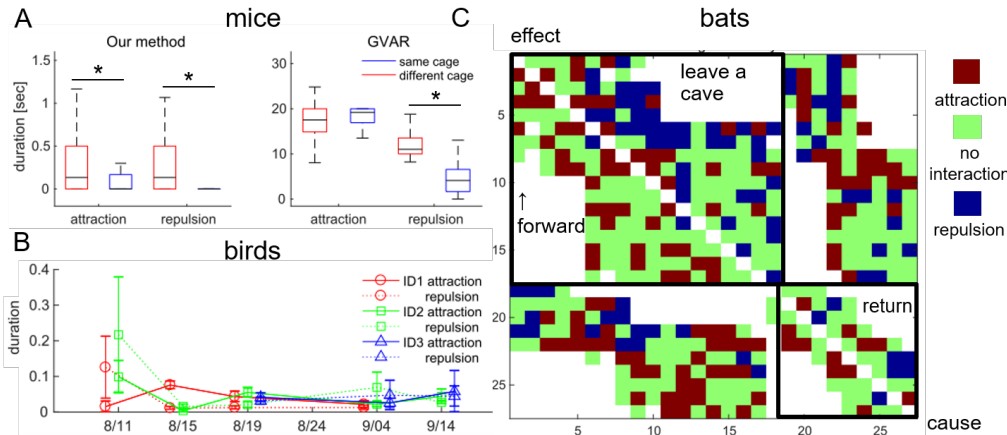

Figure 2: Analyzed results for multiple species of multi-animal trajectories. The results and details are given in the main text and Appendix H. Asterisks mean the statistically significant difference between groups ($p < 0.05$). (A) Results of three mice data grown in the different (red) and same (blue) cages. The vertical axis indicates the duration [sec] of their attraction and repulsion during 10-second bins of three interactions. Our method significantly extracted distinctive differences between cages in both movements. (B) Results of the longitudinal two or three birds data. The horizontal axis indicates the measurement date. The GPS trajectories of identified three young brown boobies (red, green, and blue) were analyzed (missing values indicates no measurement). The vertical axis indicates the normalized duration of positive (attraction: solid line) and negative (repulsion: negative) GCs for each bird (i.e., worked as the cause of another one or two birds). Error bar is the standard error among the segment during the movement. (C) Results of the observational 27 bats. The horizontal and vertical axes are the agents of the cause and effect in GC inferred by our method, respectively. The agents were sorted in the order they framed out by leaving and returning to the cave (the groups of leaving and returning were separated). The colors are the signed maximal values of the absolute GC coefficients inferred by our method, i..e, red, green, and blue indicate attraction (1), no interaction (0), and repulsion (-1), respectively.

our methods as analytical tools can obtain new insights from birds and bats datasets based on the quantitative results. Compared with the methodologies mentioned in Section 5 (i.e., uninterpretable information-theoretic approaches or using non-causal features), our method has advantages for providing local interactions: interpretable signs of Granger-causal effects over time (i.e., our findings are all new).

**Mice for verification.** As an application to a hypothesis-driven study, for example, we show the effectiveness of our method using three mice raised in different environments. The hypothesis is that, as is well known (e.g., [52]), when grown in different cages, they are more socially novel to others, thus more frequently attractive and repulsive movements will be observed. In this experiment, we regarded the same/different cage as a group pseudo-label, and confirmed that our method could extract its features in an unsupervised manner. We analyzed the trajectories of three mice in the same and different cages measured at 30 Hz for 5 min each (see also Appendix H). As shown in Figure 2A, our method extracted a significantly larger duration in the different cage than that in the same cages for both movements ($p < 0.05$; $p$ is a statistical $p$-value), whereas GVAR [45] did not in repulsive movements ($p > 0.05$) but did in attractive movements ($p < 0.05$) and extracted too much interaction. The main reason for the too much interaction in GVAR was the overdetection of the attraction and repulsion as shown in Figure 1 right (black break line). The statistical analysis and videos are presented in Appendix H and the supplementary materials. Our methods characterized the movement behaviors before the contacts with others, which have been previously evaluated (e.g., [74]).

**Growing birds.** Animals grow while interacting with other individuals, but the directed interaction between young individuals has not been fully investigated as longitudinal (i.e., long period) studies. Here, as an example, we analyzed the flight GPS (two-dimensional) trajectories of three juvenile brown boobies *Sula leucogaster* over six times for 34 days ($11.91 \pm 0.09$ [h] for each day), which were recorded at 1 Hz. We segmented two or three bird trajectories within 1 km and during moving (over 1 km/s) each other, in which interactions were considered to exist, and obtained 25 sequences of length $367 \pm 278$ [s] (for details, see Appendix H). Results of inferring GC in Figure 2B show that on the first measurement day, the most frequent directed interactions were observed between ID 1 and 2 (particularly two individuals had more repulsions and ID 1 had fewer attractions). On the

other hand, in the second and subsequent measurements, it was observed that the most interacting individuals changed every measurement day. One possible factor of the decrease in the duration of interactions (especially repulsion) may be the habituation with the same individuals. Measurements and analyses over longer periods will reveal the acquisition of social behavior in young individuals.

**Wild bats.** As an example of an exploratory analysis, we applied our method to three-dimensional trajectories of eastern bent-wing bats *Miniopterus fuliginosus* that left a cave (some bats returned to the cave). Although some multi-animal studies have investigated leader-follower relationships (see also Section 5), those in wild bats are unknown. We used two sequences with 7 and 27 bats of length 237 and 296 frames, respectively, which were obtained via digitizing the videos at 30 Hz. Details of the dataset are given in Appendix H. As a result, among 138 interactions of all 34 individuals within the leaving and returning groups, there were 46 interactions where the locationally-leading (i.e., flying forward) bats repelled the following bats in the same direction, 27 interactions where the leading bats were attracted from the following bats, 65 ones with no interactions (the results of the following bats were not discussed because it was obvious; see also example results of 27 bats in Figure 2C). Since bats can echo-locate other bats in all directions up to a range of approximately 20 m [44, 6], the locationally-leading bats can be influenced by the locationally-following bats in the same direction (if no perception, they cannot be influenced). The results suggest that the groups of flying bats would not show simple leader-follower relationships.

# 7  Conclusions

We proposed a framework for learning GC from multi-animal trajectories via a theory-based ABM with interpretable neural models. In our framework, as shown in Figure 1, the duration of interaction and non-interaction, attraction and repulsion, their amplitudes (or strength), and their timings can be interpretable. In the experiments, our method can analyze the biological movement sequence of mice, birds, and bats, and obtained novel biological insights. One possible future research direction is to incorporate other scientific knowledge into the models such as body inertia (or visuo-motor delay). Real-world animals have certain visuo-motor delays, but they also predict the others' movements (i.e., the visuo-motor delays may be smaller). This is an inherently ill-posed and challenging problem, which will be our future work.

For societal impact, our method can be utilized as real-world multi-agent analyses to estimate interaction rules such as in animals, pedestrians, vehicles, and athletes in sports. On the other hand, there are some concerns in our method from the perspectives of negative impact when applied to human data. One is a privacy problem by the tracking of groups of individuals to detect their activities and potential interactions over time. This topic has been discussed such as in [60]. Although we did not apply our method to human data, solutions for such a problem will improve the applicability of the proposed method in our society.

### Acknowledgments

This work was supported by JSPS KAKENHI (Grant Numbers 19H04941, 20H04075, 16H06541, 25281056, 21H05296, 18H03786, 21H05295, 19H04939, JP18H03287, 19H04940, and 21H05300), JST PRESTO (JPMJPR20CA), and JST CREST (JPMJCR1913). For obtaining flies data, we would like to thank Ryota Nishimura at Nagoya University.

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
