where $\boldsymbol{x}_{\leq t}^j = (..., \boldsymbol{x}_{t-1}^j, \boldsymbol{x}_t^j)$ denotes the present and past of series $j$ and $\varepsilon_t^j$ represents independent noise. We then consider that variable $\boldsymbol{x}^i$ does not Granger-cause variable $\boldsymbol{x}^j$, denoted as $\boldsymbol{x}^i \nrightarrow \boldsymbol{x}^j$, if and only if $g_j(\cdot)$ is constant in $\boldsymbol{x}_{\leq t}^i$. Granger causal relations are equivalent to causal relations in the underlying directed acyclic graph if all relevant variables are observed and no instantaneous (i.e., connections between two variables at the same timestep) connections exist [59]. Many methods for Granger causal discovery, including vector autoregressive [27] and recent deep learning-based approaches [73, 31, 81], can be encapsulated by the following framework. First, we define a function $f_\theta$ (e.g., an multilayer perceptrons (MLP) in [73], a linear model in [27]), which learns to predict the next time-step of the test sequence $\boldsymbol{x}$. Then, we fit $f_\theta$ to $\boldsymbol{x}$ by minimizing some loss (e.g., mean squared error) $\mathcal{L}$: $\theta_\star = \mathrm{argmin}_\theta \mathcal{L}(\boldsymbol{x}, f_\theta)$. Finally, we apply some fixed function $h$ (e.g., thresholding) (e.g., [45]) to the learned parameters to produce a Granger causal graph estimate for $\boldsymbol{x}$: $\hat{\mathcal{G}}_{\boldsymbol{x}} = h(\theta_\star)$.

Furthermore, we need to differentiate between positive and negative Granger-causal effects (e.g, approaching and separating). Based on the definition of [45], we define the effect sign as follows: if $g_j(\cdot)$ is increasing in all $\boldsymbol{x}^i_{\leq t}$, then we say that variable $\boldsymbol{x}^i$ has a positive effect on $\boldsymbol{x}^j$, if $g_j(\cdot)$ is decreasing in $\boldsymbol{x}^i_{\leq t}$, then $\boldsymbol{x}^i$ has a negative effect on $\boldsymbol{x}^j$. Note that $\boldsymbol{x}^i$ can contribute both positively and negatively to the future of $\boldsymbol{x}^j$ at different delays.

Overall, the causality measures, however elaborate in construction, are simply statistics estimated from a model [71]. If the model inadequately represents the system properties of interest, subsequent analyses based on the model will fail to address the question of interest. The inability of the model to represent key features of interest can cause interpretational problems and sometimes erroneous assessments of causality. Therefore, in our case, incorporating the structures of the generative process for animal behaviors (i.e., Eq.(2)) in a numerically computable form will be required. We thus propose the ABM based on a well-known conceptual model [54] in biological sciences in the next section.

## 3 Augmented behavioral model

Our motivation for developing interpretable behavior models is to obtain new insights from the results of Granger causality. In this section, we firstly formulate a well-known conceptual behavioral model [54] so that it can be computable. Second, we propose (multi-animal) ABMs with theory-based architectures based on scientific knowledge. Further, we discuss the relation to the existing explainable neural models [2]. The diagram of our method is described in Appendix C.

### 3.1 Formulation of a conceptual behavioral model

In movement ecology, which is a branch of biology concerning the spatial and temporal patterns of behaviors of organisms, a coherent framework [54] has been conceptualized to explore the causes, mechanisms, and patterns of movement. For example, two alternative structural representations [54] were proposed to model a new position $\boldsymbol{p}_{t+1}$ from its current location $\boldsymbol{p}_t$ (for details, see Appendix A): the motion-driven case $\boldsymbol{p}_{t+1} = f_U(f_M(\boldsymbol{\Omega}, f_N(\boldsymbol{\Phi}, \boldsymbol{r}_t, \boldsymbol{w}_t, \boldsymbol{p}_t), \boldsymbol{r}_t, \boldsymbol{w}_t, \boldsymbol{p}_t)) + \boldsymbol{\varepsilon}_t$, and the navigation-driven case $\boldsymbol{p}_{t+1} = f_U(f_N(\boldsymbol{\Phi}, f_M(\boldsymbol{\Omega}, \boldsymbol{r}_t, \boldsymbol{w}_t, \boldsymbol{p}_t), \boldsymbol{r}_t, \boldsymbol{w}_t, \boldsymbol{p}_t)) + \boldsymbol{\varepsilon}_t$, where $\boldsymbol{w}_t$ is the internal state, $\boldsymbol{\Omega}$ is the motion capacity, $\boldsymbol{\Phi}$ is the navigation capacity, and $\boldsymbol{r}_t$ is the environmental factors (these are conceptual parameters). $f_M$, $f_N$, and $f_U$ are conceptual functions to represent actions of the motion (or planning), navigation, and movement progression processes, respectively.

For efficient learning of the weights in the model (i.e., coefficient of Granger causality) in this paper, we consider a simple case with homogeneous navigation and motion capacities, and internal states. Moreover, to make the contribution of $f_M$, $f_N$, and $f_U$ interpretable after training from the data for extracting unknown interaction rules (and for obtaining scientific new insights), one of the simplified processes for agent $i$ is represented by

$$\boldsymbol{x}^i_{t+1} = f^i_U(f^i_N(\boldsymbol{r}^i_t, \boldsymbol{x}^i_t), f^i_M(\boldsymbol{r}^i_t, \boldsymbol{x}^i_t), \boldsymbol{r}^i_t, \boldsymbol{x}^i_t) + \boldsymbol{\varepsilon}^i_t, \tag{2}$$

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

## A  Nathan's conceptual framework for movement ecology

Based on [54], we can model the movement of an organism from its current location $\boldsymbol{p}_t$ to a potentially new position $\boldsymbol{p}_{t+1}$, as a function of its current location $\boldsymbol{p}_t$, internal state $\boldsymbol{w}_t$, motion capacity $\boldsymbol{\Omega}$, navigation capacity $\boldsymbol{\Phi}$, and their interactions with the current environmental factors $\boldsymbol{r}_t$. This implies a general relationship

$$\boldsymbol{p}_{t+1} = F(\boldsymbol{\Omega}, \boldsymbol{\Phi}, \boldsymbol{r}_t, \boldsymbol{w}_t, \boldsymbol{p}_t). \tag{8}$$

The insight comes from being as specific as possible about the structure of $F$, without sacrificing framework generality. Using the notation $f_M$, $f_N$, and $f_U$ to represent actions of the motion, navigation, and movement progression processes, respectively, [54] posited two alternative structural representations, the motion-driven case

$$\boldsymbol{p}_{t+1} = f_U(f_M(\boldsymbol{\Omega}, f_N(\boldsymbol{\Phi}, \boldsymbol{r}_t, \boldsymbol{w}_t, \boldsymbol{p}_t), \boldsymbol{r}_t, \boldsymbol{w}_t, \boldsymbol{p}_t)) + \boldsymbol{\varepsilon}_t, \tag{9}$$

and the navigation-driven case

$$\boldsymbol{p}_{t+1} = f_U(f_N(\boldsymbol{\Phi}, f_M(\boldsymbol{\Omega}, \boldsymbol{r}_t, \boldsymbol{w}_t, \boldsymbol{p}_t), \boldsymbol{r}_t, \boldsymbol{w}_t, \boldsymbol{p}_t)) + \boldsymbol{\varepsilon}_t. \tag{10}$$

In the motion-driven case, the navigation process can be viewed as creating a map of probabilities for the locations to which the individual can potentially move at time $t + 1$. The motion process weights these probabilities, thereby altering their relative values. In the navigation-driven case, the navigation process depends on how $\boldsymbol{w}_t$, $\boldsymbol{r}_t$, and $\boldsymbol{u}_t$ interact with the motion process and $\boldsymbol{\Phi}$ to enable navigation. Indeed, some organisms may alternate between the two types of movement; however, in both cases, the movement progression process $f_U$ evaluates the weighted probabilities presented by the potential movement map, thereby determining the next position.

For efficient learning to use only one time series in most (neural) GC frameworks [73, 31, 45], in this paper, we consider the simple case with homogeneous navigation and motion capacities, and internal states. Moreover, to make the contribution of $f_M$, $f_N$, and $f_U$ interpretable after training from the data without assuming either motion-driven or navigation-driven case, one of the simplified processes for agent $i$ is represented by

$$\boldsymbol{x}_{t+1}^i = f_U^i(f_N^i(\boldsymbol{r}_t^i, \boldsymbol{x}_t^i) f_M^i(\boldsymbol{r}_t^i, \boldsymbol{x}_t^i), \boldsymbol{r}_t^i, \boldsymbol{x}_t^i) + \boldsymbol{\varepsilon}_t^i, \tag{11}$$

where $\boldsymbol{x}^i$ includes location $\boldsymbol{u}^i$ and velocity for agent $i$. This equation is same as Eq. (2).

## B  Self-explaining neural networks

Self-explaining neural networks (SENNs) were introduced [2] as a class of intrinsically interpretable models motivated by explicitness, faithfulness, and stability properties. A SENN with a link function $g(\cdot)$ and interpretable basis concepts $h(x) : \mathbb{R}^p \to \mathbb{R}^k$ is expressed as follows:

$$f(\boldsymbol{x}) = g(\theta(\boldsymbol{x})_1 h(\boldsymbol{x})_1, ..., \theta(\boldsymbol{x})_k h(\boldsymbol{x})_k), \tag{12}$$

where $\boldsymbol{x} \in \mathbb{R}p$ are predictors; and $\theta(\cdot)$ is a neural network with k outputs. We refer to $\theta(\boldsymbol{x})$ as generalized coefficients for data point $\boldsymbol{x}$ and use them to *explain* contributions of individual basis concepts to predictions. In the case of $g(\cdot)$ being sum and concepts being raw inputs, Eq. (4) simplifies to $f(\boldsymbol{x}) = \sum_{j=1}^p \theta(\boldsymbol{x})_j \boldsymbol{x}_j$. Appendix B presents additional properties SENNs need to satisfy and the learning algorithm, as defined by [2]. The SENN was first applied to GC [45] via GVAR such that

$$\boldsymbol{x}_t = \sum_{k=1}^K \boldsymbol{\Psi}_{\boldsymbol{\theta}_k}(\boldsymbol{x}_{t-k}) \boldsymbol{x}_{t-k} + \boldsymbol{\varepsilon}_t, \tag{13}$$

where $\boldsymbol{\Psi}_{\boldsymbol{\theta}_k} : \mathbb{R}^p \to \mathbb{R}^{p \times p}$ is a neural network parameterized by $\boldsymbol{\theta}_k$. For brevity, we omit the intercept term here and in the following equations. No specific distributional assumptions are made on the noise terms $\boldsymbol{\varepsilon}_t$. $\boldsymbol{\Psi}_{\boldsymbol{\theta}_k}(\boldsymbol{x}_{t-k})$ is a matrix whose components correspond to the generalized coefficients for lag $k$ at timestep $t$. In particular, the component $(i, j)$ of $\boldsymbol{\Psi}_{\boldsymbol{\theta}_k}(\boldsymbol{x}_{t-k})$ corresponds to the influence of $\boldsymbol{x}_{t-k}^j$ on $\boldsymbol{x}_t^i$.

As defined by [2], $g(\cdot)$, $\theta(\cdot)$, and $h(\cdot)$ in Equation 2 need to satisfy:

1) $g$ is monotone and completely additively separable

2) For every $z_i := \theta_i(x) h_i(x)$, $g$ satisfies $\frac{\partial g}{\partial z_i} \geq 0$

3) $\theta$ is locally difference bounded by $h$

4) $h_i(x)$ is an interpretable representation of $x$

5) $k$ is small.

A SENN is trained by minimizing the following gradient-regularized loss function, which balances performance with interpretability: $\mathcal{L}_y(f(\boldsymbol{x}), y) + \lambda \mathcal{L}_{\boldsymbol{\theta}}(f(\boldsymbol{x}))$, where $\mathcal{L}_y(f(\boldsymbol{x}), y)$ is a loss term for the ground classification or regression task; $\lambda > 0$ is a regularization parameter; and $\mathcal{L}_{\boldsymbol{\theta}}(f(\boldsymbol{x})) = \|\nabla_{\boldsymbol{x}} f(\boldsymbol{x}) - \boldsymbol{\theta}(\boldsymbol{x})^\top J_{\boldsymbol{x}}^h(\boldsymbol{x})\|_2$ is the gradient penalty, where $J_{\boldsymbol{x}}^h$ is the Jacobian of $h(\cdot)$ w.r.t. $\boldsymbol{x}$. This penalty encourages $f(\cdot)$ to be locally linear.

In this paper, we utilized a SENN approach as augmented theory-based models for flexible and interpretable modeling with the following regularization utilizing scientific knowledge. Note that we actually used a quasi-SENN which does not always satisfy 3) ($\theta$ is locally difference bounded by $h$) in Appendix B because biological movement sequences are inherently time-varying dynamics and do not need the temporal stability required in [45].

## C    Overview of our method

The overview of our algorithm is simple as shown in Figure C.1. In Section 3.2, we formulate ABM. ABM is learnt in Section 4.1 with the theory-guided regularization described in Section 4.2. The model is described in Eq. (5) and the objective function is Eq. (6). Finally, using the obtained coefficient $Psi_t$, the Granger causality is inferred in Section 4.3.

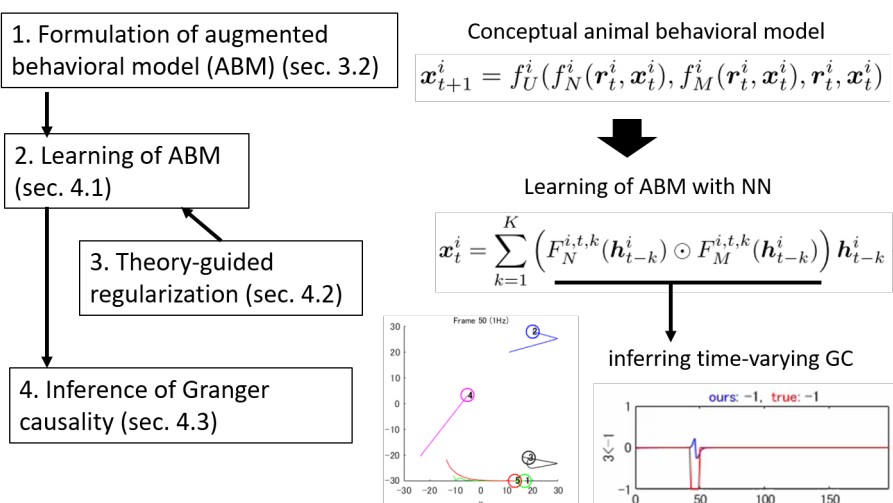

Figure C.1:  Block diagram of our method.

## D    Common training setup

### D.1    Model training and the amount of computation

This experiment was performed on an Intel(R) Xeon(R) CPU E5-2699 v4 (2.20 GHz $\times$ 16) with GeForce TITAN X pascal GPU. For the training of the proposed and baseline models [45], we used the Adam optimizer [32] with an initial learning rate of 0.0001 and 500 training epochs. The learning rate was decayed by a factor of 0.995 for each epoch. We set the batchsize to the time length of sequences $T - K - 1$. The hidden layer is a two-layer MLPs of size 50.

In addition, to compare the methods in terms of their amount of computation, we measured training and inference time across two datasets. We eliminated ACD [43] due to the completely different framework, and linear GC and local TE had obviously shorter computation time due to their simple

architectures. The results in Table D.3 show that the computation time of our method was between eSRU [31] and GVAR [45] for both datasets. That is, since our method requires a larger input dimension than GVAR [45], our method took a higher computational cost than GVAR [45], but it was more efficient than eSRU [31].

Although we performed experiments on relatively small datasets, we can estimate the computation time for larger datasets. Similarly to most of the Granger causality methods, we computed the Granger causality for each sequence, thus the computation time is linear with respect to the number of sequences.

| | Kuramoto model $(p = 5, T = 200, K = 5)$ | Boid model $(p = 5, T = 200, K = 3)$ |
|---|---|---|
| eSRU [31] | $162 \pm 5$ | $143 \pm 9$ |
| GVAR [45] | $27 \pm 3$ | $19 \pm 1$ |
| ABM (full) | $116 \pm 4$ | $129 \pm 4$ |

Table D.3: The averaged computation time [s] among 10 sequences in two datasets.

### D.2 Inference of GC in our model with the binary threshold

To infer a binary matrix of GC relationships in our method, we use a heuristic threshold. GVAR [45] proposed a stability-based procedure that relies on time-reversed GC (TRGC) [80], which proved the validity of time reversal for linear finite-order autoregressive processes. However, since our problem includes time-varying nonlinear dynamics, our method cannot leverage the TRGC framework. In our work, we use a heuristic threshold $\max_{K+1 \leq t \leq T} \left( \max_{1 \leq k \leq K} \left| \mathbf{\Psi}'_{i,j} \right|_F \right) /2$, because the values of the GC matrix $S_{i,j}$ vary for each sequence due to the learning framework. We assume approximately $1 : 1$ with and without GC in all experiments, but in other cases, if more or less case, we need to modify the threshold. If possible, we can examine it using a validation dataset.

### D.3 Baseline models implementation

We compared the performances of our method to infer GC with those in the following baselines using two synthetic datasets. Except for ACD [43], most baselines did not provide an obvious way for handling multi-dimensional time series, whereas our main problems include two- or three-dimensional trajectories for each animal. Therefore, we modified the baselines except for ACD to compute norms with respect to spatial dimensions (2 or 3) for comparability with the proposed method. For Kuramoto datasets, the input of ACD and eSRU was a two-dimensional vector concatenating $\frac{d\phi_i}{dt}$ and the intrinsic frequencies $\omega_i$), and that of linear GC and local TE was $\frac{d\phi_i}{dt}$ (one-dimension).

**eSRU** [31]. This approach based on economic statistical recurrent units is an extension of an original neural GC method [73] using MLPs and LSTMs. We performed grid search in sparsity hyperparameters $\lambda_1 \in [0.01, 0.05]$, $\lambda_2 \in [0.01, 0.05]$, $\lambda_3 \in [0.01, 0.1]$ according to [45]. Based on the performances in various experiments in [31], the number of layers in the second stage of the feedback network was set to 2 and the Adam optimizer was used with an initial learning rate of $0.001$ and $2,000$ training epochs (for other hyperparameters we used default values). The same threshold for a binary matrix of GC relationships was used in our method. We used the implementation in https://github.com/sakhanna/SRU_for_GCI.

**ACD** [43]. This approach is based on the neural relational inference (NRI) [33] for the Granger-causal discovery using graph neural networks and variational autoencoders. This approach requires no hyperparameter optimization for model training, but used training sequences for pre-training of the model. For fair comparisons, we used 10 training sequences for the pre-training and then performed test-time adaptation for the test dataset. For both synthetic experiments, the latent dimension throughout the model was set to size 64 due to the small dataset size. The remaining hyperparameters were the same as the default values of the previous work [33]. For example, we optimized the model using the Adam optimizer with a learning rate of 0.0005. We trained the model for 500 epochs. We used the implementation in https://github.com/loeweX/AmortizedCausalDiscovery.

**GVAR** [45]. This is our base model without scientific knowledge, and the implementation details are given in Appendix D.1. We used a stability-based procedure that relies on the TRGC described above. We used the implementation in https://openreview.net/forum?id=DEa4JdMWRHp.

**Linear GC and Local TE**. First, we computed a linear version of GC, where non-zero linear weights are taken as greater causal importance. Second, we computed TE at each timestep as the local TE, which has been used in many biological researches. The same threshold for a binary matrix of linear GC relationships was used as our method. We used the implementation of [81] in `https://github.com/tailintalent/causal`.

# E  Kuramoto model and the augmented model

## E.1  Simulation model

The Kuramoto model is a nonlinear system of phase-coupled oscillators that can exhibit a range of complicated dynamics based on the distribution of the oscillators' internal frequencies and their coupling strengths. We use the common form for the Kuramoto model given by the following differential equation:

$$\frac{d\phi_i}{dt} = \omega_i + \sum_{j \neq i} k_{ij} \sin(\phi_i - \phi_j) \tag{14}$$

with phases $\phi_i$, coupling constants $k_{ij}$, and intrinsic frequencies $\omega_i$. We simulate one-dimensional trajectories by solving Eq. (14) with a fourth-order Runge-Kutta integrator with a step size of $0.01$.

We simulate $5$ phase-coupled one-dimensional oscillators with intrinsic frequencies $\omega_i$ and initial phases $\phi_i^{t=1}$ sampled uniformly from $[1, 10)$ and $[0, 2\pi)$, respectively. We randomly, with a probability of $0.5$, connect pairs of oscillators $v_i$ and $v_j$ (undirected) with a coupling constant $k_{ij} = 1$. All other coupling constants were set to $0$.

## E.2  Augmented model

Here, we describe the specific form of Eq. (3) for the Kuramoto model. We did not use the navigation function, i.e., we regard the perception process $f_N$ as an identity map.

To avoid overfitting and model selection problem, we simply design the function $F_M^i$ and the input features $\boldsymbol{h}_M^{i,j}$ based on Eq. (14). The output of Eq. (3) is the differential value of the one-dimensional phase. The function value $F_M^i(\boldsymbol{h}_M^{i,j}) \in \mathbb{R}^d$ representing coefficients of the self and other elements information is computed by the following procedure. Based on Eq. (14), we design the interpretable feature $\boldsymbol{h}_M^{i,j}$ by concatenating $\frac{d\phi_i}{dt}$ and $\sin(\phi_i - \phi_j)$ for all $j \neq i$, and the intrinsic frequencies $\omega_i$ (copied for every timestep as $\omega_i$ are static). The function $F_M^i(\cdot)$ is implemented by the two-layer MLPs for each $k$ and element $i$ with $(p+1)$-dimensional input and $1$-dimensional output.

The theory-guided regularization is similar to that of the animal model (i.e., considering only no interaction case), due to the difficulty in the prediction of integral error of the fourth-order Runge-Kutta. The hyperparameters in Eq. (6) were determined by the grid search of $\lambda \in [0, 0.1]$, $\beta \in [0, 0.025]$, and $\gamma \in [0.1, 10000]$. The order $K$ was set to $5$ which was the same as [45] for not time-varying dynamics.

# F  Results of the Kuramoto model

Here, we validated our method on the Kuramoto dataset, which contains five time-series of phase-coupled oscillators [35]. This is because it has been still difficult to detect GC without a large amount of data [43] rather than other synthetic datasets indicating higher detection performance such as in [31, 45]. For our base augmented model, see Appendix E.

The results are shown in Table F.3, indicating that our method achieved much better performance than various baselines.

|           | Kuramoto model | | | |
|-----------|-------------------|-------------------|-------------------|-------------------|
|           | Acc. | Bal. Acc. | AUROC | AUPRC |
| Linear GC | $0.655 \pm 0.099$ | $0.500 \pm 0.000$ | $0.546 \pm 0.139$ | $0.431 \pm 0.143$ |
| Local TE  | $0.335 \pm 0.107$ | $0.483 \pm 0.050$ | $0.489 \pm 0.054$ | $0.351 \pm 0.104$ |
| eSRU [31] | $0.500 \pm 0.092$ | $0.500 \pm 0.000$ | $0.487 \pm 0.123$ | $0.548 \pm 0.121$ |
| ACD [43]  | $0.475 \pm 0.121$ | $0.528 \pm 0.115$ | $0.605 \pm 0.135$ | $0.519 \pm 0.184$ |
| GVAR [45] | $0.495 \pm 0.154$ | $0.473 \pm 0.113$ | $0.467 \pm 0.079$ | $0.398 \pm 0.115$ |
| ABM - $\mathcal{L}_{TG}$ | $\mathbf{0.930 \pm 0.075}$ | $\mathbf{0.914 \pm 0.086}$ | $\mathbf{0.972 \pm 0.036}$ | $\mathbf{0.929 \pm 0.093}$ |
| ABM (full) | $0.925 \pm 0.075$ | $0.902 \pm 0.098$ | $\mathbf{0.972 \pm 0.036}$ | $\mathbf{0.929 \pm 0.093}$ |

Table F.3: Performance comparison on the Kuramoto model. Standard deviations (SD) are evaluated across 10 replicates.

# G    Boid model and the augmented model

## G.1    Simulation model

The rule-based models represented by time-varying dynamical systems have been used to generate generic simulated flocking agents called boids [62]. The schooling model we used in this study was a unit-vector-based (rule-based) model [12], which accounts for the relative positions and direction vectors neighboring fish agents, such that each fish tends to align its own direction vector with those of its neighbors. In this model, 5 agents (length: 0.5 m) are described by a two-dimensional vector with a constant velocity (1 m/s) in a boundary square ($30 \times 30$ m) as follows: $\boldsymbol{r}^i = (x_i \ y_i)^T$ and $\boldsymbol{v}_t^i = \|\boldsymbol{v}^i\|_2 \boldsymbol{d}_i$, where $x_i$ and $y_i$ are two-dimensional Cartesian coordinates, $\boldsymbol{v}^i$ is a velocity vector, $\|\cdot\|_2$ is the Euclidean norm, and $\boldsymbol{d}_i$ is an unit directional vector for agent $i$.

At each timestep, a member will change direction according to the positions of all other members. The space around an individual is divided into three zones with each modifying the unit vector of the velocity. The first region, called the repulsion zone with radius $r_r = 1$ m, corresponds to the "personal" space of the particle. Individuals within each other's repulsion zones will try to avoid each other by swimming in opposite directions. The second region is called the orientation zone, in which members try to move in the same direction (radius $r_o$). We set $r_o = 2$ to generate swarming behaviors. The third is the attractive zone (radius $r_a = 8$ m), in which agents move towards each other and tend to cluster, while any agents beyond that radius have no influence. Let $\lambda_r$, $\lambda_o$, and $\lambda_a$ be the numbers in the zones of repulsion, orientation and attraction respectively. For $\lambda_r \neq 0$, the unit vector of an individual at each timestep $\tau$ is given by:

$$\boldsymbol{d}_i(t + \tau, \lambda_r \neq 0) = - \left( \frac{1}{\lambda_r - 1} \sum_{j \neq i}^{\lambda_r} \frac{\boldsymbol{r}_t^{ij}}{\|\boldsymbol{r}_t^{ij}\|_2} \right), \tag{15}$$

where $\boldsymbol{r}^{ij} = \boldsymbol{r}_j - \boldsymbol{r}_i$. The velocity vector points away from neighbors within this zone to prevent collisions. This zone is given the highest priority; if and only if $\lambda_r = 0$, the remaining zones are considered. The unit vector in this case is given by:

$$\boldsymbol{d}_i(t + \tau, \lambda_r = 0) = \frac{1}{2} \left( \frac{1}{\lambda_o} \sum_{j=1}^{\lambda_o} \boldsymbol{d}_j(t) + \frac{1}{\lambda_a - 1} \sum_{j \neq i}^{\lambda_a} \frac{\boldsymbol{r}_t^{ij}}{\|\boldsymbol{r}_t^{ij}\|_2} \right). \tag{16}$$

The first term corresponds to the orientation zone while the second term corresponds to the attraction zone. The above equation contains a factor of $1/2$ which normalizes the unit vector in the case where both zones have non-zero neighbors. If no agents are found near any zone, the individual maintains a constant velocity at each timestep.

In addition to the above, we constrain the angle by which a member can change its unit vector at each timestep to a maximum of $\beta = 30$ deg. This condition was imposed to facilitate rigid body dynamics. Because we assumed point-like members, all information about the physical dimensions of the actual fish is lost, which leaves the unit vector free to rotate at any angle. In reality, however, the conservation of angular momentum will limit the ability of the fish to turn angle $\theta$ as follows:

$$\boldsymbol{d}_i(t + \tau) \cdot \boldsymbol{d}_i(t) = \begin{cases} \cos(\beta) & \text{if } \theta > \beta \\ \cos(\theta) & \text{otherwise.} \end{cases} \tag{17}$$

If the above condition is not unsatisfied, the angle of the desired direction at the next timestep is rescaled to $\theta = \beta$. In this way, any un-physical behavior such as having a $180°$ rotation of the velocity vector in a single timestep is prevented.

### G.1.1   Simulation procedure

The initial conditions were set such that the particles would generate a torus motion, though all three motions emerge from the same initial conditions. The initial positions of the particles were arranged using a uniformly random number on a circle with a uniformly random radius between 6 and 16 m (the original point is the center of the circle). The average values of the control parameter $r_o$ were in general 2, 10, and 13 to generate the swarm, torus, and parallel behavioral shapes, respectively. In this paper, in average, we set $r_o = 2$ and $r_a = 8$, and $r_r = 1$ in attractive relationship and $r_r = 10$ in repulsive relationship. We simply added noise to the constant velocities and the above three parameters among the agents (but constant within the agent) with a standard deviation of $0.2$. We finally simulated ten trials in 2 s intervals (200 frames). The timestep in the simulation was set to $10^{-2}$ s.

### G.2   Augmented model

Here, we describe the specific form of Eq. (3). To avoid overfitting and model selection problems, we simply design the functions $F_N^i$ and $F_M^i$ and the input features $h_N^{i,j}$ and $h_M^{i,j}$ based on Eqs. (15), (16), and (17). The output of Eq. (3) is limited to the velocity because the boid model does not depend on the self-location and involves the equations regarding velocity direction. The boid model assumes constant velocity for all agents, but our augmented model does not have the assumption because the model output is the velocity, rather than the velocity direction.

First, the navigation function value $F_N^i(h_N^{i,j}) \in \mathbb{R}^{p-1}$ representing the signed information for other agents is computed by the following procedure (for simplicity, here we omitted the time index $t$ and $k$). We simply design the interpretable features $h_N^{i,j}$ by concatenating $v^{i,j}$ and $\|r^{i,j}\|_2$ for all $j \neq i$, where $v^{i,j}$ is the velocity of agent $i$ in the direction of $r^{i,j}$ (i.e., if agent $i$ approaches $j$ like Eq. (16), $v^{i,j}$ is positive, and if separating from $j$, $v^{i,j}$ is negative like Eq. (15)). The specific form of $F_N^i(h_N^{i,j})$ is

$$F_N^i(h_N^{i,j}) = \varsigma_{a_d}\left(\frac{1}{\|r^{i,j}\|_2} - d_{ignore}\right)\left(\varsigma_{a_v}(v^{i,j}) - \frac{1}{2}\right) \times 2, \tag{18}$$

where $\varsigma_{a_d}, \varsigma_{a_v}$ are sigmoid functions with gains $a_d, a_v$, respectively, and $d_{ignore}$ is a threshold for ignoring other agents. $(\varsigma_{a_v}(v^{i,j}) - 1/2) \times 2$ represents the signs of effects of $j$ on $i$, where the value is positive if agent $i$ is approaching to $j$ like Eq. (16), and it is negative if separating from $j$ like Eq. (15). we set $a_v = 1e - 2$. $\varsigma_{a_d}(1/\|r^{i,j}\|_2)$ represents whether the agent $i$ ignores $j$ or not and is zero if the agents $i, j$ are infinitely far apart. For $d_{ignore}$, if we assume that all agents can see other agents in the analyzed area, we set $d_{ignore} = 0$ and $a_d = 1e - 6$ (birds and mice datasets in our experiments). Otherwise, we set $a_d = 1e - 2$ and $d_{ignore} \in \mathbb{R}^1$ can be estimated via the back-propagation using the loss function in Eq. (6).

Next, the movement function value $F_M^i(h_M^{i,j}) \in \mathbb{R}^d$ representing coefficients of the self and other agents information is computed by the following procedure. Based on Eqs. (15), (16), and (17), we design the interpretable feature $h_M^{i,j}$ by concatenating $v^i \in \mathbb{R}^d$ and $r^{i,j}/\|r^{i,j}\|_2 \in \mathbb{R}^d$ for all $j \neq i$. The movement function $F_M^i(\cdot)$ is implemented by the two-layer MLPs for each $k$ and agent $i$ with $dp$-dimensional input and $d$-dimensional output.

The hyperparameters in Eq. (6) were determined by the grid search of $\lambda \in [0.01, 1000]$, $\beta \in [0, 0.025]$, and $\gamma \in [1, 10000]$. The order $K$ was set to 3 because it would be difficult to model the time-varying dynamics by using too large $K$.

## H   Multi-animal trajectory data and experiments

In this section, we describe the details of animal datasets and the results. Videos are given in the supplementary materials. For all statistical calculations, $p < 0.05$ was considered as significant.

### H.1 Mice.

We analyzed the 5-min trajectories of groups of three mice raised in the same or different cages (C57BL6J; 1 year old; male or female) voluntarily walking in an open arena (55 cm × 60 cm). Experiments were conducted in accordance with Doshisha University Institutional Animal Care and Use Committee. The two-dimensional coordinates of snout, nape, and tail base were obtained from images captured at 30 frames per second, using a USB digital video camera mounted 1.3 m above the open arena via an image tracking software, *DeepLabCut* [48]. We used the averaged values of all estimated joint coordinates for the subsequent analysis. We evaluated the duration of the interaction using the threshold described in Appendix D.2 for every 10 sec with no overlap (i.e., we evaluated $N = 30$ sequences). To compare the interaction duration between groups, the Kruskal-Wallis test was used because most of the data did not follow normal distributions using the Lilliefors test. As the post-hoc comparison, the Wilcoxon rank sum test with Bonferroni correction was used within the factor where a significant effect in Kruskal-Wallis test was found. We used $r$ values as the effect size for Wilcoxon rank sum test. Our method extracted significantly distinctive differences between the cages in both movements ($p < 0.033, r > 0.27$), whereas GVAR [45] did not in repulsive movements ($p > 0.05$) but did in attractive movements ($p < 0.001, r = 0.75$) and extracted too much interaction (10 [s] indicates three mice interacted during $1/3$ of all duration).

### H.2 Birds.

We analyzed the flight GPS trajectories of three juvenile brown boobies *Sula leucogaster* over six times for 34 days in 2010 ($11.91 \pm 0.09$ [h] for each day), which were recorded at 1 Hz. Some authors raised three brown booby chicks of unknown sexes. After fledging, animal-borne GPS loggers were attached to the backs of juvenile brown boobies. The measurement was conducted under the approval of the Nature Conservation Division in Okinawa, Japan (see [82], for methodological detail). We segmented two or three bird trajectories within 1 km and during moving (over 1 km/s) each other, in which interactions were considered to exist, and obtained 25 sequences of length $367 \pm 278$ [s]. Table H.3 indicates a more detailed characteristics of the birds dataset.

| Date (in 2010) | 8/11 | 8/15 | 8/19 | 8/24 | 9/4 | 9/14 |
|---|---|---|---|---|---|---|
| Recording [hours] | 12.01 | 11.93 | 11.92 | 12.00 | 11.78 | 11.87 |
| No. of sequences | 2 | 2 | 7 | 0 | 12 | 2 |
| Time length [sec] | $187 \pm 74$ | $487 \pm 119$ | $256 \pm 136$ | N/A | $452 \pm 363$ | $310 \pm 103$ |

Table H.3: Characteristics of the birds dataset.

### H.3 Bats.

We analyzed three-dimensional trajectories of eastern bent-wing bats *Miniopterus fuliginosus*. This species mainly inhabits in caves. In such a cave, females begin to gather just before breeding, beginning at the end of June, and breeding care was conducted (for details, see [20]). This cave, utilized as a breeding cave, was reported to shelter approximately 20,000 bats. Each evening during the breeding period, around sunset, bats emerged from the cave in groups. In this study, measurements were taken in front of the cave at approximately 19:00 on July 12 and 15, 2019. We used two sequences obtained via digitizing the videos at 30 Hz (videos were recorded at 60 Hz). The direct linear transformation method was used to estimate the 3D position coordinates calculated via camera calibration from known 3D coordinates (calibration points) of images obtained from the two cameras. One included 7 bats interactions of length 237 frames (a bat left the cave and 6 bats returned to the cave). Another included 27 bats interactions of length 296 frames (17 bats left the cave and 10 bats returned to the cave). We analyzed GC inferred by our method within each group. The agents were sorted in the order they framed out by leaving and returning to the cave (the groups of leaving and returning were separated) as shown in Figure 2. Moreover, we did not quantitatively analyze the GC results of (locationally) backward bats against the forward bats, because the backward bats obviously followed the forward bats. Thus, the number of the analyzed interaction was computed such that $_6C_2 + {}_{17}C_2 + {}_{10}C_2 = 196$. However, the behaviors of each bat frame-in and frame-out in the digitized area, thus we finally obtained 138 interactions of all 34 bats.

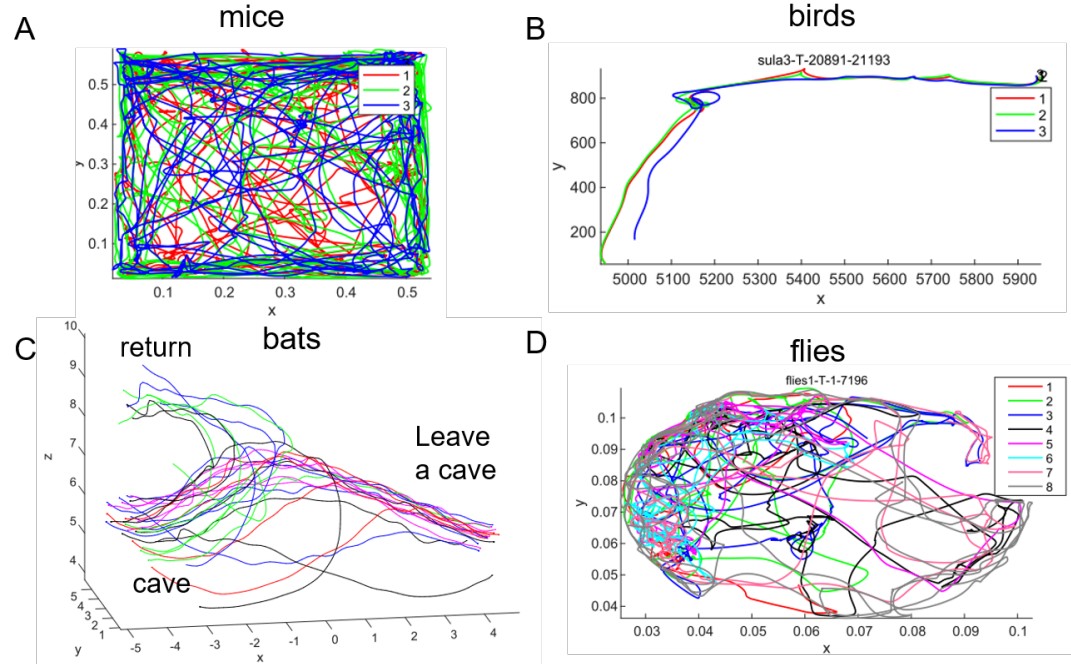

Figure H.4: Example trajectories in each animal dataset. The units of all axes are meters. (A) Three mice in 5 minutes. (B) Three birds (right is the starting point) in 302 seconds. (C) 27 bats (lower left is the starting point) in 9.87 seconds. (D) Eight male flies in 4 minutes.

## H.4 Flies.

Similarly to the mice dataset, as an application to a hypothesis-driven study, we show the effectiveness of our method using eight flies in different female-male ratios. Male flies actively pursue females, but do not pursue other males, as is well known (e.g., [14]). Based on this knowledge, we hypothesized that males are more socially novel to others including female flies (called mixed group) than the male-only group, thus more frequently attractive and repulsive movements will be observed in the mixed group. In this experiment, we regarded the mixed/male-only group as a group pseudo-label, and confirmed that our method can extract its features in an unsupervised manner. Canton-S strain was used as a wild-type of *Drosophila melanogaster*. Flies were raised on standard cornmeal yeast medium at $25 \pm 1$ °C and 40%–60% relative humidity in 12 h/12 h light/dark cycle. Males and females were collected during 24 h after eclosion. Males were maintained in isolation until experiments. Females were maintained in a group with males until experiments. Eight flies (6-8 days old) were applied into a chamber with modified size (11.4cm diameter) from [68] for video recording. We analyzed the trajectories of 8 flies in the mixed (4 males and females) and male-only (8 males) groups measured at 30 Hz for 4 min each. The two-dimensional coordinates were obtained via Ctrax [7]. We evaluated the duration of the interaction fly using the threshold described in Appendix D.2 for every 10 seconds with no overlap (i.e., we evaluated $N = 24$ sequences). Since the numbers of male flies were different in both groups, we computed the interaction duration for each male.

To compare the interaction duration between groups, we used the same statistical test (the Mann-Whitney U-test) as the mice dataset. As shown in Figure H.4, our method and GVAR [45] extracted significantly larger duration in the mixed group than that in the male-only group for both movements ($p < 0.039, r > 0.28$), whereas GVAR extracted too much interaction ($10[sec]$ indicates flies interacted during $1/7$ of all duration). See also the videos given in the supplementary materials to confirm the fewer interactions than those estimated by GVAR. In summary, our method characterized the male flies' social behaviors as attractive and repulsive movements.

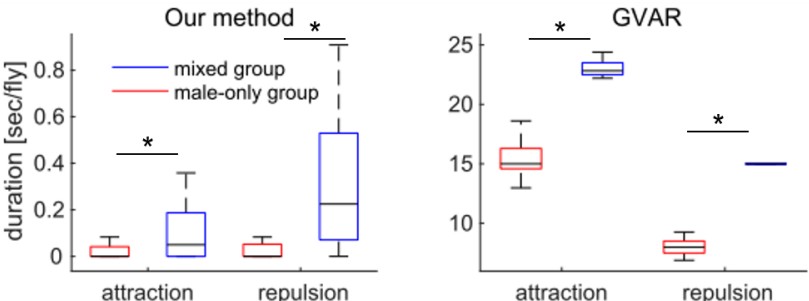

Figure H.4: Results of eight flies data in the mixed (red) and only-male (blue) groups. The vertical axis indicates the duration [sec/fly] of their attraction and repulsion during 10-second bins of seven interactions for each fly (i.e., the maximum duration was 70 seconds).