# OpenReview forum: "Learning interaction rules from multi-animal trajectories via augmented behavioral models"
_NeurIPS.cc/2021/Conference — NeurIPS 2021 Poster_

### Official Review · Reviewer_C9GF · 2021-07-11

**Rating:** 7
**Confidence:** 3

**Summary:**

The proposed paper introduces a novel approach for learning the Granger causality based on augmented behavioral models. The objective of the discussed method is to extract interaction rules from real-world multi-agent trajectory data with neural networks. The main contributions of this work are a framework for learning Granger causality, methods for efficient and interpretable learning, and the decomposition of motion and navigation. The method is evaluated based on a number of experiments and based on datasets of mice, flies, birds, and bats.


**Ethical Concerns:**

I do not have any ethical or diversity concerns toward the topic this paper addresses or the paper in general.


**Limitations And Societal Impact:**

Limitations of the method are not discussed. In the Conclusion the authors suggest that their method could have negative impact, but the discussion remains rather vague. Both should be prepared appropriately prior to publishing this work.

**Main Review:**

Overall, this is an interesting paper that has most things going for it. The proposed framework is technically-sound, the experiments are well thought-out, and the results are promising. Furthermore, the topic the paper addresses is important and I appreciate the interdisciplinary character of this work. However, the paper is somewhat dense and many sections are not easy to comprehend. I have the impression that more care could be taken in providing additional details and explanations across Section 3 and 4. Especially, the motivation for developing interpretable behavior models comes too short and interpretability of the proposed framework is not assessed. Moreover, several sentences show grammar and spelling mistakes that need to be taken care of (examples provided below). So in conclusion, I think this paper could be accepted if proper care is taken in preparing the final version of this work. However, the paper would also benefit from a more thorough revision outside the review cycle from NeurIPS. Therefore, I am not fully too enthusiastic about accepting this work.

Specific comments:

- L1: Is 'moving sequences' really the correct meaning here? Or should it be 'movement sequences'? Similar in L407 it doesn't seem 'moving sequence' is the right grammar.

- L27, L83: For the less movement-ecology-versed reader it would be helpful to provide more information about what is meant by 'generative process in biological multi-agent trajectories' or the 'generative process in animal behaviors.

- L88: What is meant by 'we incorporate the structures into the base model of a CG framework'? Sentences like this are too vague and need to be clarified. What is meant by 'structures'?

- In Sections 3 and 4, I am missing a clearer description of the model and what components are actually being learned. Perhaps an overview figure would help to make this more clear.

- L70 'an multi layer perceptrons' --> 'a ... perceptron'

- L71 and L72: 'some loss', 'some fixed function'. This sounds too vague and it would be helpful to be more specific here.

- L103: A clearer description as to why only one time series is considered needs to be provided here. It is also not clear based on the description provided in Line 653 as it is mostly a copy from the main text.

- L105: While I understand the motivation to make the contributions of f_M, f_N, and f_U interpretable -- which is in fact an interesting objective -- it appears somewhat out of context and more details are necessary. As is the paragraph (L103-112) is too difficult to understand. Similar in Line 120. What is the goal for making f_* interpretable? What this goal accomplished?

- L131-132, L134-135: These sentences read too repetitive.

- In Section 4.1. it is not clear what is actually learned. What is the model architecture that is being trained?

- L346: 'were sensitive the sign as shown' --> rephrase.

-L649: If i am not mistaken u_t is not being referenced anywhere.

**Time Spent Reviewing:**

4

---

> ### Author Response · Authors · 2021-08-10
> **We thank you for your constructive comments.**
>
> We address the questions/concerns below. We will fix typos in the revised paper.
>
> 1. Sections 3 and 4: The motivation for developing interpretable behavior models is to obtain new insights from the results of Granger causality with the time-varying sign indicating attraction and repulsion. The interpretability of our framework was evaluated in the boid dataset (Figure 1) with the ground truth of the interaction. We will add these points to the revised paper.
>
> 2. Moving sequences: We appreciate your advice. As you pointed out, 'movement sequences' may be better. We will modify it in the revised paper.
>
> 3. Generative process (L27, L83): it means Eq. (2) (but it is conceptual and not numerically computable). We will explicitly associate the generative process and Eq. (2) in the revised paper.
>
> 4. Structure (L88): These are the structures of the generative process in Eq. (2) mentioned in L87. Eq (2) is conceptual, but we formulated this in Eq. (3) in an implementable form.
>
> 5. Clearer description of the learned model: This is related to Eq. (3) and the above L143-149. We consider the creation of a figure which helps the understanding of them.
>
> 6. 'some loss', 'some fixed function' (L71 and L72): We will add examples of them regarding previous and our work. For example, we can consider mean squared error as 'some loss'.
> As 'some fixed function, we can consider stability-based thresholding [Marcinkevics et al. 2020, ICLR] (but it did not work well in our study) and our heuristic threshold in Appendix C.1.
>
> 7. Why only one time series is considered (L103): In the original and most neural Granger causality frameworks, weights in the model (or coefficient of Granger causality) are learnt for each time series, rather than batches of time series. This characteristic was not directly linked to the following sentences, but just a premise. Since this may be confusing in our submitted paper, we will reconstruct this sentence more clearly.
>
> 8. Difficulty in understanding (L103-112) and the goal for making f_* interpretable: For the former (L103-112), we will revise this paper so that it becomes easier to understand by a figure with explanations which helps the understanding of them. For the latter, the goal for making the functions f_* interpretable is to extract unknown interaction rules based on knowledge of biological sciences for obtaining scientific new insights. We accomplished new biological insights using our framework in birds and bats datasets described in Section 6.2.
>
> 8. Repetition in L131-132, L134-135: We will delete the former.
>
> 9. The actually learned parameter in Section 4.1: Various Fs and Psis in Eq. (5) are learned parameters.
>
> 10. u_t in L649 (appendix). Sorry, this should be p_t instead of u_t.
>
> Limitation: In conclusion, we mentioned that we need to incorporate other scientific knowledge such as body inertia (or visuo-motor delay) for deeper scientific understanding. In particular, in the boid and real-world datasets, we assume that agents can use information one to three timesteps before (i.e., the order K of the autoregressive model, but the sampling frequency depends on the datasets). Real-world animals have certain visuo-motor delays, but they also predict the others' movements (i.e., the visuo-motor delays may be smaller). This is an inherently ill-posed and challenging problem, which will be our future work. We will add this point in the revised paper.
>
> Societal Impact: In the revised paper, as another reviewer pointed out, in application to human datasets, we need to consider potential problems regarding the tracking of multi-person activities such as privacy problems. We will add the ethical concerns in the revised paper.

---

> > ### Comment · Reviewer_C9GF · 2021-09-03
> > **Thank you**
> >
> > Thank you for your respones. These clarifications would be essential if the paper were to be accepted.

---

> > > ### Author Response · Authors · 2021-09-03
> > > **Many thanks**
> > >
> > > We appreciate your feedback and increasing your score. Of course, we would like to revise the paper to improve the clarification.

---

### Official Review · Reviewer_gmyy · 2021-07-13

**Rating:** 6
**Confidence:** 3

**Summary:**

In this paper, the authors propose a method for learning rules of interaction between individuals from　trajectory data of animals. Their method is based on the Granger causality methodology for time series data but differs from previous studies in that it introduces a new network architecture and a regularization term based on the theory of animal behavior, which enables more accurate learning. The method has been validated using synthetic datasets and then applied to real animal behavior data to gain new knowledge about animal behavior.

**Limitations And Societal Impact:**

Limitations are not written by the authors, but it is not clear whether the method can be applied to large datasets. There seems to be no negative impact on society.

**Main Review:**

### Strengths

* In-depth and extensive experiments. After validating the method on two types of synthetic data, the authors apply the method to real animal movement trajectory data. In the real data experiments, the authors first check whether the results of the proposed method are consistent with existing knowledge on mice data, and then apply the method to birds and bats data to gain new knowledge. All of these experiments have a clear purpose and work well to present the advantages and features of the proposed method.

* The datasets used in the experiment. The animal trajectory datasets are collected for this paper, which is rare in the machine learning community. It is interesting to see the new insights derived by the proposed method from such data.

* This paper is well-written; the problems, the backgrounds, and the contributions are clearly stated with appropriate citations.

### Weaknesses

* Small novelty of the method. It is claimed that the novelty of the method is the introduction of a network architecture and a regularization term based on the theory of animal behavior to the existing Granger causality estimation method, but this in itself is a straightforward combination of previous methods and has little originality.

* Validation has only been done on datasets where the problem size is relatively small. It would be of great interest to analyze whether it is possible to successfully find Granger causality on larger datasets, what the computation time would be, and whether any other problems would arise.

### Other comments
* Are the discoveries about growing birds and wild bats new discoveries or are they a validation of previous discoveries?  In particular, for wild bats, it would be good to compare and discuss with the contents of [57].

* This paper may possibly be preferred more at data mining conferences such as KDD than at machine learning conferences such as NeurIPS.

**Time Spent Reviewing:**

5

---

> ### Author Response · Authors · 2021-08-10
> **We thank you for your constructive comments.**
>
> We address the questions/concerns below.
>
> 1. Our methodological contribution: We consider that the novelty of our method is beyond the combination of existing methods. First, we realized the theory-guided regularization for reliable biological behavioral modeling for the first time. The theory-guided regularization can leverage scientific knowledge such that “when this situation occurs, it would be like this” (i.e., domain experts often know an input and output pair of the prediction model). Existing methods in Granger causality did not consider the utilization of such knowledge. Second, biologically, a well-known conceptual behavioral model [Nathan et al. 2008, PNAS] did not have a numerically computable form. One of our methodological contributions in biology also lies in the reformulation of this model such that we can compute and quantitatively evaluate it. We will add these points in the revised paper.
>
> 2. Dataset size: although we performed experiments on relatively small datasets, we can estimate computation time for larger datasets. Similarly to most of the granger causality methods, we computed the Granger causality for each sequence, thus the computation time is linear with respect to the number of sequences.
>
> 3. New discoveries about growing birds and wild bats: We obtained new biological insights using our framework via discussion with authors who are experts in each animal, as described in each paragraph of Section 6.2. As a premise, please note that biological studies have never used interpretable information-theoretic or Granger causality analysis (e.g., the time-varying sign indicating attraction and repulsion; see also the second paragraph in Section 5) and analyzed using non-causal features (e.g., cross-correlation). Therefore our findings were all new. In the birds dataset, directed interactions decreased as the measurement date progressed possibly because of the habituation with the same individuals. In the bats dataset, the locationally-leading bats can be inﬂuenced by the locationally-following bats in the same direction, suggesting that the groups of ﬂying bats would not show simple leader-follower relationships. In such a way, the proposed framework can provide new types of insights which would be difficult to be obtained by conventional biological approaches. Therefore, it is expected to lead to new biological discoveries in multi-animal behaviors.
>
> 4. Limitation: In conclusion, we mentioned that we need to incorporate other scientific knowledge such as body inertia (or visuo-motor delay) for deeper scientific understanding.  In particular, in the boid and real-world datasets, we assume that agents can use information one to three timesteps before (i.e., the order K of the autoregressive model, but the sampling frequency depends on the datasets). Real-world animals have certain visuo-motor delays, but they also predict the others' movements (i.e., the visuo-motor delays may be smaller). This is an inherently ill-posed and challenging problem, which will be our future work. We will add this point in the revised paper.
>
> 5. Negative impact on society: We mentioned this in the conclusion section. Additionally, according to another reviewer's advice, in application to human datasets, we need to consider potential problems regarding the tracking of multi-person activities such as privacy problems. We will add the ethical concerns in the revised paper.

---

> ### Author Response · Authors · 2021-09-03
> **Dear reviewer gmyy**
>
> We appreciate your time and the very informative comments, and we would like to inquire whether our response addressed your concerns. If you found our response convincing, we kindly ask you to increase your score.

---

### Official Review · Reviewer_hrQV · 2021-07-13

**Rating:** 6
**Confidence:** 4

**Summary:**

The manuscript proposes a hybrid method for learning Granger causality that combines data-driven learning methods and theory-based behavioral models for the goal of modeling multi-agent behavior.

The main motivation is that current models either do not consider the data generation process behind the observed time series or if they do, they are tailored to specific animal species, thus having generalization limitations.

More specifically, a Self-Explainable Neural Network (SENN) -like model is used where scientific knowledge on multi-agent interactions is integrated. In this regard, key contributions of the proposed method include: a theory-guided loss function to regularize training and a motion and navigation function Experiments on synthetic and realistic data show the effectiveness of the proposed method.

**Ethical Concerns:**


Regarding potential negative applications of the proposed method. The proposed method has a clear negative behind tracking groups of individuals over time, be it in video recordings or via their activities and potential interactions in social networks. In this regards, the manuscript limits itself to provide a very superficial pointer towards fairness issues when operating on top of data from individuals.
While I do not find a potential dual use issue a barrier to stop the publication of a good idea, I do believe that these issues should be at least acknowledged in a proper manner. I would encourage the authors, in the appendix,  to discuss at length potential problems regarding the tracking of collective activities in social networks.

**Ethics Review Area:**

["Inappropriate Potential Applications & Impact  (e.g., human rights concerns)"]

**Limitations And Societal Impact:**


Regarding potential negative applications of the proposed method. The proposed method has a clear negative behind tracking groups of individuals over time, be it in video recordings or via their activities and potential interactions in social networks. In this regards, the manuscript limits itself to provide a very superficial pointer towards fairness issues when operating on top of data from individuals.
While I do not find a potential dual use issue a barrier to stop the publication of a good idea, I do believe that these issues should be at least acknowledged in a proper manner. I would encourage the authors, in the appendix,  to discuss at length potential problems regarding the tracking of collective activities in social networks.

**Main Review:**

The proposed method is sound and different design decisions behind are properly motivated. The content of the manuscript is relatively clear and has a good flow.

In addition, the proposed method was tested on real-world datasets, where it proved effective at validating observations already made in other fields.

My main criticisms with the manuscript are the following:


- C1: The text in l.118-119 gives the impression that models that learn parameters or rules in a purely data-driven manner are inherently not interpretable. I suggest rephrasing this sentence to avoid this confusion since  this this statement is clearly not accurate.


- C2: The introduction of the manuscript and presentation of the proposed method were properly motivated and detailed. While these are desirable properties for a manuscript, those two parts are quite extended when compared to the experimental part of the manuscript. Relevant aspects of the conducted experiments and results are delegated to the Appendix. For instance, the ablation study in Appendix G shows the effect that the proposed theory-guided regularization $L_{TG}$ and learning navigation $F^k_N$ and motion $F^k_M$ functions (the main contributions of the manuscript) have on performance. This is a critical experiment that should be in the main body of the paper.
Due to delegations like this, the manuscript does not feel as a self contained document.

- C3: In l.311-313  it is stated that for the baselines it is not obvious how to handle multi-dimensional time series
In the one hand, the capability of handling multi-dimensional data it is a strength of the your method. I would suggest emphasizing this strength in the manuscript.
On the other hand, the observation (l.311-313) makes me wonder whether the selected baselines are the most appropriate ones for a fair comparison with respect to the proposed method.

- C4: For the experiments in the real multi-animal trajectory datasets, in l.358-359 it is stated that since there is no ground truth for real-world data, the hyperparameters of the boid simulation dataset are re-used.
I was wondering, how applicable is the re-use of these  parameters when going from a simplistic synthetic setting to a more realistic one? Are there potential side-effects that could be expected? Did you observe any problem, possibly caused by this factor, during the execution of your experiments on real-world data?

- C5: In several parts of the manuscript it is stated that through the proposed method novel biological insights were obtained. I was wondering were these novel insights validated by experts in the respective fields?

**Post Rebuttal Update**
I thank the author for addressing my review on their rebuttal. Having read all the reviews and the feedback provided by the authors, this is my current position with respect to the manuscript:

Overall the authors addressed to a good extent most of the weak points and criticisms raised on my review, which will be included in a revised version of the submitted manuscripts.

I agree with reviewer C9GF on that some parts of the manuscript could use some extra attention in order to improve their presentation.

The ethical reviewers share my opinion on that while the proposed method poses some potential risks, this are not strong enough to reject it. More importantly, these are aspects that could be easily addressed with a proper discussion either in the manuscript or in its appendix.

On the optimistic side, I believe the manuscript has the potential to be better given that all the amendments promised in the rebuttal are applied in the revised version. On the pessimistic side, I am not sure whether it would be possible to include all the possible amendments given the space limitations imposed in the manuscript.

Keeping the previous points, I am in favor of increasing initial rating towards "6: Marginally above the acceptance threshold" but will not push for the acceptance of the manuscript. In this regard, I am more in favor of trying to improve the current weak aspects of the manuscript rather than trying to get it accepted in its current form. (This observation seems to be shared by C9GF)



**Needs Ethics Review:**

Yes

**Time Spent Reviewing:**

4

---

> ### Author Response · Authors · 2021-08-10
> **We thank you for your constructive comments.**
>
> We address the questions/concerns below.
>
> C1. As you pointed out, the sentence was inaccurate. We will revise the sentence such that  “in purely data-driven manners (sometimes uninterpretable)”.
>
> C2. We appreciate your advice. As you commented, the ablation results in Appendix G are important as numerical experiments. Although we prioritized biological results in real-world multi-animal data, we will add these abrasion results in the revised paper.
>
> C3. We thank you for the advice. We will emphasize the strength of handling multi-dimensional data. As a baseline, since our model basically extended GVAR [Marcinkevics et al. 2020, ICLR], the most appropriate baseline is considered to be GVAR. We will clarify this point in the revised paper.
>
> C4. That is a good point. Although we did not find that our analysis caused serious problems in the application to real-world datasets (in particular, for mice and flies, we had clear hypotheses and the results supported them), we did not fully explain the potential side-effects. A potential side-effect is to infer the Granger causality inaccurately due to body inertia (or visuo-motor delay) as mentioned in the conclusion section. In the boid and real-world datasets, we assume that agents can use information one to three timesteps before (i.e., the order K of the autoregressive model, but the sampling frequency depends on the datasets). Real-world animals have certain visuo-motor delays, but they also predict the others' movements (i.e., the visuo-motor delays may be smaller). This is an inherently ill-posed and challenging problem, which will be our future work. We will add these points in the revised paper.
>
> C5. In the birds and bats datasets, we obtained new biological insights using our framework via discussion with authors who are experts in each animal, as described in each paragraph of Section 6.2. As a premise, please note that biological studies have never used interpretable information-theoretic or Granger causality analysis (e.g., the time-varying sign indicating attraction and repulsion; see also the second paragraph in Section 5) and analyzed using non-causal features (e.g., a cross-correlation between two agents). Therefore our findings are all new. In the birds dataset, directed interactions decreased as the measurement date progressed possibly because of the habituation with the same individuals. In the bats dataset, the locationally-leading bats can be inﬂuenced by the locationally-following bats in the same direction, suggesting that the groups of ﬂying bats would not show simple leader-follower relationships. In such a way, the proposed framework can provide new types of insights which would be difficult to be obtained by conventional biological approaches. Therefore, it is expected to lead to new biological discoveries in multi-animal behaviors.
>
> Ethical Concerns: In this paper, we focused on multi-animal behaviors, but as you pointed out, in application to human datasets, we need to consider potential problems regarding the tracking of multi-person activities such as privacy problems. We will add the related ethical concerns in the revised paper.

---

> ### Author Response · Authors · 2021-09-03
> **Many thanks**
>
> We appreciate your post rebuttal feedback and increasing your score. We would like to revise the paper to address your concerns.

---

### Official Review · Reviewer_cyHu · 2021-07-16

**Rating:** 6
**Confidence:** 3

**Summary:**

This paper proposes a framework for learning Granger causality with the goal of finding interaction rules from trajectory data of multiple animals. The framework has an "augmented behavior model" based on a conceptual behavior model studied in [53]. The experiments are on a range of datasets, from synthetic to real data of mice, birds, and bats. The main baseline of comparison is GVAR [44], which this method compares against for synthetic data and on real mice data. Based on the experiments, the interaction rules found are in the form of attraction, no interaction, or repulsion between different agents.

**Limitations And Societal Impact:**

I didn't see where the authors mentioned the limitations of their method - would be great if the authors could point me in the right direction. There is a paragraph in line 411 addressing societal impacts.

**Main Review:**

1.  I appreciate that the authors tested on a range of datasets. However, I feel the results method could be more clear. For example, in Figure 2A, the authors say that "our method extracted a significantly larger duration in the same cage than that in the different cages", but the figure shows that the duration of same cage is lower for both attractive and repulsion, so this point is not clear to me. The y-axis between this method and GVAR is also significantly different - can the authors comment on this? Also, what does the "*" mean in Figure 2A? Additionally, the authors mention that the model is interpretable (ex: lines 8, 10), but there are many ways to interpret whether a model is interpretable and I'm not sure where interpretability is demonstrated in the experiments (is it by the duration of interaction & non-interaction?). Could the authors clarify this point?
2. Overall, I found the methods section difficult to understand. Perhaps a diagram or pseudo-code of the algorithm (ex, as provided by GVAR [44] in Algorithm 1) might enable the authors to more clearly communicate their setup to a broader range of audiences who may not be familiar with this sub-field.
3.  Line 291: "for application to multi-animal trajectory datasets that can only be measured in small quantities". Why can multi-animal trajectory data only be in small quantities? Some labs are releasing large amounts of trajectory data collected over time. I agree that it is difficult and time-consuming to measure this but I don't agree that these datasets can only be in small quantities.

Post Rebuttal Update:

I've read the other reviews and the author response, and my current rating is raised to borderline, learning towards accept (assuming the authors will apply all the changes promised in the rebuttal). This is because the authors have clarified some of my points of confusion with the paper (methods as well as experiments). I still feel the paper could be presented more clearly, but the authors have mentioned adding a figure or a pseudo-code of their algorithm, which I feel would improve clarity of the methods section substantially. I agree with reviewer hrQV that if all the amendments in the rebuttal are applied to the paper, the revision would be stronger.



**Time Spent Reviewing:**

4

---

> ### Author Response · Authors · 2021-08-10
> **We thank you for the constructive comments**
>
> We address the questions/concerns below.
>
> 1. Figure 2A: We appreciate your pointing out. Our hypothesis and Figure 2A were correct, but the sentence was incorrect. Correctly, "our method extracted a significantly larger duration in the different cage than that in the same cages". The main reason for too much interaction in GVAR was the overdetection of the attraction and repulsion as shown in Figure 1 left (black break line). Asterisks (*) mean the statistically significant difference between groups (p < 0.05). We will improve the clarity in the revised paper.
>
> 2. Interpretability: We can interpret when and how agents interact. Compared with classical Granger causality methods, we can consider that our and previous frameworks [Marcinkevics et al. 2020, ICLR] are more interpretable. In our framework, in addition to what you suggest (the duration of interaction and non-interaction), as shown in Figure 1, attraction and repulsion (the sign of the y-axis), their amplitudes (or strength), and their timings are also interpretable. We can rigorously evaluate the results of such variables in a synthetic data experiment (such as Figure 1) with ground truth, but in the real-world data, we have no ground truth of such variables. Therefore, we compared the thresholded results (Figure 2C) and their duration (Figures 2A and 2B) for two or three groups.
>
> 3. Diagram or pseudo-code of the algorithm: We thank you for the advice. Although we will add a pseudo-code or diagram of our algorithm in the revised paper, the overview of our algorithm is simple. In summary, as described in Section 4, the model is described in Eq. (5) and the objective function is Eq. (6). Using the obtained coefficient Psi_t, the Granger causality is inferred in Section 4.3.
>
> 4. Amounts of multi-animal trajectory data: We did not intend that multi-animal trajectory datasets are always measured in small quantities. Correctly, we should revise the sentence such that  ".. would not be suitable for application to the multi-animal trajectory datasets that are measured in small quantities".
>
> 5. Limitation: In conclusion, we mentioned that we need to incorporate other scientific knowledge such as body inertia (or visuo-motor delay) for deeper scientific understanding. In particular, in the boid and real-world datasets, we assume that agents can use information one to three timesteps before (i.e., the order K of the autoregressive model, but the sampling frequency depends on the datasets). Real-world animals have certain visuo-motor delays, but they also predict the others' movements (i.e., the visuo-motor delays may be smaller). This is an inherently ill-posed and challenging problem, which will be our future work. We will add this point in the revised paper.

---

> ### Author Response · Authors · 2021-09-03
> **Dear reviewer cyHu**
>
> We appreciate your time and very informative comments, and we would like to inquire whether our response addressed your concerns. If you found our response convincing, we kindly ask you to increase your score.

---

> > ### Author Response · Authors · 2021-09-04
> > **Many thanks**
> >
> > We appreciate your post rebuttal feedback and increasing your score. We would like to revise the paper according to all reviewers' comments.

---

### Review · Ethics_Reviewer_yeWm · 2021-07-23

**Recommendation:**

I would encourage the authors to expand their societal impacts statement, as the reviewer suggests and as described above.

**Ethics Review:**

I did not review the technical aspects of this paper, just the ethical issues. (This is not part of my review, but I will note that I have wondered for a long time whether my cat follows me around, or I follow him around, and this paper appears to offer an answer to this question, so I at least am personally interested in their research question! Kudos to the authors.)

3/4 reviewers do not raise any ethics concerns. The fourth notes that "The proposed method has a clear negative behind tracking groups of individuals over time, be it in video recordings or via their activities and potential interactions in social networks. In this regards, the manuscript limits itself to provide a very superficial pointer towards fairness issues when operating on top of data from individuals. While I do not find a potential dual use issue a barrier to stop the publication of a good idea, I do believe that these issues should be at least acknowledged in a proper manner. I would encourage the authors, in the appendix, to discuss at length potential problems regarding the tracking of collective activities in social networks."

I do not believe this paper has sufficiently severe ethical issues that it should be rejected. I largely agree with the reviewer, however, and would encourage the authors to put more care into their societal impact statement (the last paragraph of the paper). I agree that the potential negative applications in unethical mass surveillance systems merit mention, as the original reviewer suggests. However, because this concern basically applies to any method which analyzes human interactions, I don't think it's sufficiently severe that the paper should be rejected. I also think that the discussion of fairness in the conclusion "One is ignoring a fairness problem, i.e., it may include bias e.g., when applied to human data" is very superficial and should be expanded to a sentence or two with relevant citations.

---

> ### Author Response · Authors · 2021-09-03
> **We thank you for the constructive comments**
>
> We appreciate your ethical review. We would like to revise the paper to address the related surveillance and fairness problems.

---

### Review · Ethics_Reviewer_TpJj · 2021-08-11

**Recommendation:**

It is possible to address the concerns mentioned above by spending a bit more time discussing the negative impacts of surveillance and privacy related to the techniques addressed in the paper.

**Ethical Issues:**

Yes

**Ethics Review:**

This paper borders on the issues of surveillance without directly addressing surveillance or privacy. The authors mentioned there maybe "bias" in this approach when it is used on real humans and identified this as a negative societal impact, however there was no further elaboration included. For the paper to be applied to humans there would be a need for surveillance. The purpose of the technique seems to designed to interpret complex behaviors of animal and/or human interaction. Some complex behaviors, or their causes, maybe private, but with this technique they may be exposed with or without the consent of those affected. This paper can do more to address these issues in more depth.

---

> ### Author Response · Authors · 2021-09-03
> **We thank you for the constructive comments**
>
> We appreciate your ethical review. We would like to revise the paper to address the related surveillance and privacy problems.

---

### Decision · Program_Chairs · 2021-09-27

**Decision:**

Accept (Poster)

**Comment:**

The paper proposes a hybrid method for learning Granger causality that combines data-driven learning methods and theory-based behavioral models for the goal of modeling multi-agent behavior. The paper addresses an important problem and can stimulate further research in this direction.
Initial reviews were split on the technical contribution and also the technique being a combination of existing methods.
However, after the discussion and rebuttal the reviewers have turned more positive about the ideas presented in the paper.
The AC echoes the view point in this case that combining existing methods is often challenging and can reveal important scientific insights.
Overall after the discussion all reviewers are in agreement about the value of the contribution and recommend acceptance.